# Regulation of cadherin dimerization by chemical fragments as a trigger to inhibit cell adhesion

Akinobu Senoo [1], Sho Ito [2,3], Satoru Nagatoishi [4 ✉], Yutaro Saito [1], Go Ueno [5], Daisuke Kuroda [1,6], Kouhei Yoshida [6], Takumi Tashima[1], Shota Kudo[1], Shinsuke Sando [1,6] & Kouhei Tsumoto [1,4,6 ✉]

Many cadherin family proteins are associated with diseases such as cancer. Since cell adhesion requires homodimerization of cadherin molecules, a small-molecule regulator of dimerization would have therapeutic potential. Herein, we describe identification of a P-cadherin-specific chemical fragment that inhibits P-cadherin-mediated cell adhesion. Although the identified molecule is a fragment compound, it binds to a cavity of P-cadherin that has not previously been targeted, indirectly prevents formation of hydrogen bonds necessary for formation of an intermediate called the X dimer and thus modulates the process of X dimerization. Our findings will impact on a strategy for regulation of protein-protein interactions and stepwise assembly of protein complexes using small molecules.

[1] Department of Chemistry and Biotechnology, Graduate School of Engineering, The University of Tokyo, Hongo, Bunkyo-ku, Tokyo, Japan. [2] Graduate School of Life Science, University of Hyogo, Kamigori-cho, Ako-gun, Hyogo, Japan. [3] ROD (Single Crystal Analysis) Group, Application Laboratories, Rigaku Corporation, Akishima, Tokyo, Japan. [4] Institute of Medical Science, The University of Tokyo, Shirokanedai, Minato-ku, Tokyo, Japan. [5] RIKEN SPring-8 Center, Sayo-cho, Sayo-gun, Hyogo, Japan. [6] Department of Bioengineering, Graduate School of Engineering, The University of Tokyo, Hongo, Bunkyo-ku, Tokyo, Japan. ✉email: ngtoishi@ims.u-tokyo.ac.jp; tsumoto@bioeng.t.u-tokyo.ac.jp

Protein assembly underlies almost all of the biological processes[1]. There has been so much reported structural information on homomers and heteromers in various kinds of functional complexes[2]. Modulators of protein assembly formation have the potential as drugs and as probes to investigate protein function. However, the fundamental interaction of the assembly formation; protein–protein interactions are difficult to regulate with small molecules for several reasons. First, large-surface areas are usually involved in interactions between proteins, whereas the accessible surface areas of chemical ligands are small. In addition, there are generally no substantial grooves at the protein–protein interface, and there are few natural inhibitors of protein–protein interactions to guide ligand design[3–5].

Cell adhesion by classical cadherin family proteins, calcium-dependent cell-adhesive molecules, entails protein assembly formation. Depending on the biological context, cadherins can play tumor-promoting roles in various tissues[6]. P-cadherin is a classical cadherin family protein. Its overexpression in some cancer tissues has been reported[7–12], and promotes metastasis and proliferation[13–18]. As the formation of cell aggregates blocks anoikis[8], and P-cadherin-mediated signaling is important for cancer cell survival[17], inhibition of P-cadherin function is a potential anticancer strategy. Cell adhesion is achieved through *trans* homodimerization between cadherins on apposed cells and then *cis* clustering of cadherins on the same cell surface[18] (Fig. 1a). For some type I classical cadherins, including P-cadherin, *trans* homodimerization has been intensively studied, and the two extracellular domains (EC12) of five on the protein that interact during the stepwise process have been identified[19–22] (Fig. 1a). The X dimer is an intermediate state that favorably promotes the final dimerized state, which is called the strand-swap dimer (S–S dimer). The S–S dimer has a unique binding mode in which a tryptophan residue of the N terminal strand of one monomer is swapped into a hydrophobic pocket of the other monomer. Based on this binding mode, several peptide mimetic ligands have been reported[23–28]; however, no ligands that bind to the hydrophobic pocket have been identified probably because the hydrophobic pocket is always occupied with either tryptophan of its own or of the other monomer at any stage in the process of S–S dimerization[21], resulting in a poor understanding of the molecular basis of inhibition of homodimerization by these peptide mimetics. Orthosteric inhibition of S–S dimerization, which is the last stage in the assembly process, may not be the best way to regulate the process. We have to change direction to find other ways to effectively regulate the cell-adhesive assembly with a small molecule.

In this study, we performed a screen for inhibitors at any stage of the process. We screened a library that contained small-molecule drug-like fragments and identified a compound that bound to a novel ligand-binding site unique to P-cadherin. The compound inhibited cell adhesion through the modulation of the X dimerization, indirectly blocking the formation of hydrogen bonds that stabilize the X dimer.

## Results

**SPR-based fragment screen.** SPR-based fragment screen was performed as described previously[29] (Fig. 1b). The fragment library was ordered from Drug Discovery Initiative; it contained 1973 compounds. As a primary screen, a direct binding assay was performed. Monomer mutant of P-cadherin called REC12[21], where arginine is inserted at the N terminus of EC12 (see

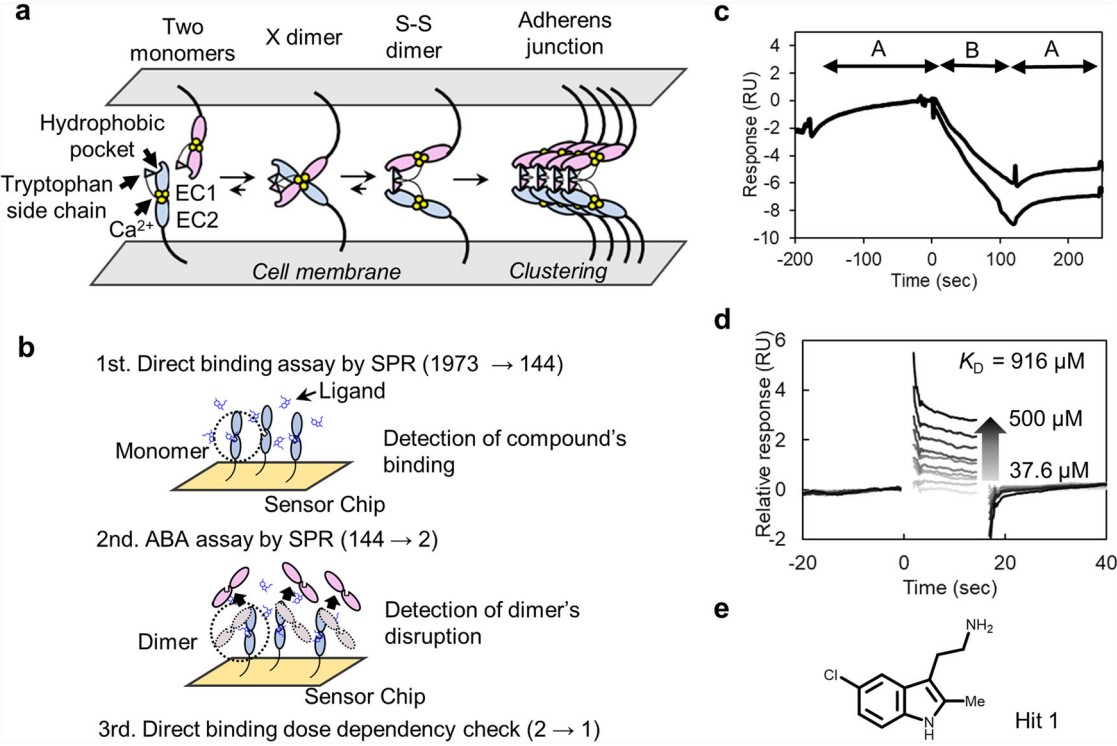

**Fig. 1 SPR-based fragment screening. a** Illustration of the stepwise dimerization of P-cadherin to mediate cell adhesion and a schematic of the protein (EC12). **b** Schematic of fragment screening. The numbers in the parentheses show decreases in the number of fragment candidates. **c** Representative sensorgram from ABA assay. EC12, in monomer-S–S dimer equilibrium, was immobilized on Sensor Chip CM5. The decrease of response upon injection of solution B shows that the compound disrupted the dimer formed on the sensor chip. **d** Dose–response analysis of binding of Hit 1 to REC12 immobilized on Sensor Chip SA. The putative $K_D$ was calculated by the Scatchard method using the SPR responses in equilibrium. Note that the $K_D$ value is not reliable, since the binding response was not saturated even at the highest concentration of Hit 1. $N = 3$. **e** Chemical structure of Hit 1.

**Table 1 Crystallographic data collection and refinement statistics.**

| Data collection | REC12-Hit 1 | MEC12-Hit 1 |
|---|---|---|
| PDB ID | 7CMF | 7CME |
| Space group | C 1 2 1 | P 2$_1$ 2$_1$ 2$_1$ |
| Unit-cell dimensions | | |
| a, b, c (Å) | 74.3, 40.8, 72.6 | 79.8, 99.1, 107.9 |
| α, β, γ (°) | 90, 97.4, 90 | 90, 90, 90 |
| Wavelength (Å) | 1.0000 | 1.0000 |
| Resolution (Å)* | 36.85–2.30 | 45.04–2.45 |
| | (2.39–2.30) | (2.60–2.45) |
| $R_{merge}$ | 0.15 (1.09) | 0.07 (1.56) |
| $R_{meas}$ | 0.19 (1.28) | 0.08 (1.67) |
| $CC_{1/2}$ | 0.99 (0.53) | 0.99 (0.54) |
| <I/σ(I)> | 7.80 (1.28) | 19.21 (1.30) |
| Completeness (%) | 94.4 (97.3) | 99.9 (99.8) |
| Redundancy | 3.1 (3.1) | 7.4 (7.5) |
| Refinement statistics | | |
| Resolution (Å) | 36.85–2.30 | 45.04–2.45 |
| $R_{work}$ | 0.245 | 0.209 |
| $R_{free}$ | 0.287 | 0.249 |
| No. of non-hydrogen atoms | 1664 | 3388 |
| Macromolecules | 1626 | 3303 |
| Ligands | 18 | 56 |
| Solvent | 20 | 29 |
| Unique reflections | 9268 (940) | 31974 (3199) |
| Average B-factor (Å$^2$) | 54.57 | 75.81 |
| Macromolecules | 54.46 | 75.33 |
| Ligands | 67.2 | 112.2 |
| MolProbity score | 2.58 | 2.42 |
| R. M. S. deviations from ideal | | |
| Bonds (Å) | 0.005 | 0.01 |
| Angles (°) | 0.77 | 1.15 |
| Ramachandran plot (%) | | |
| Favored region | 94.76 | 96.46 |
| Allowed region | 4.76 | 3.07 |
| Outlier region | 0.48 | 0.47 |

*Values in parentheses are for the highest-resolution shell.

**Identification of a ligand-binding site for Hit 1.** We employed X-ray crystallography to identify the binding site of Hit 1 to monomer, REC12 and to elucidate the inhibitory mechanism of Hit 1. REC12 was crystallized and soaked with Hit 1 at 10 mM final concentration. The resulting complex diffracted to 2.30 Å resolution (Fig. 2a, Table 1, and Supplementary Fig. S2). Electron density that can be modeled as Hit 1 was observed inside a shallow cavity located between the EC1 and EC2 domains. Upon binding of Hit 1, the side chain of Y140 shifted relative to its position in the crystal structure of REC12 alone (PDB ID; 4zmz) (Fig. 2b), and Y140 and Hit 1 form a CH−π interaction. Note that both structures with or without Hit 1 are highly isomorphic, so that the difference should not arise from the packing effect. A water molecule was located in the cavity in the crystal structure of REC12 alone (PDB ID; 4zmz). Thus, the driving forces that result in the binding of Hit 1 appear to be the interaction with Y140 residue and dehydration. Hit 1 also has some van der Waals interactions with R68, V98, T99, D100, and D137 (Fig. 2c). In order to confirm the relevant electron density observed in the crystal structure reflects the binding of Hit 1, we performed a SPR-based direct binding assay using REC12 Y140R. Hit 1 did not bind to this mutant (Supplementary Fig. S3), validating the contribution of Y140 to the binding observed in the crystal structure. Several residues around this binding pocket are reported to be important for X dimerization including Y140, D100, and Q101. In the X dimer, hydrogen bonds between Y140 and K14, and between Q101 and D100 are observed[21]. Although Hit 1 does not disrupt these hydrogen bonds directly, alternation of the structure of this region apparently inhibits X dimerization.

Interestingly, when we superposed the P-cadherin-Hit 1 complex structure with that of E-cadherin (PDB ID; 2o72), we found that the volume of the cavity in E-cadherin was smaller than that in P-cadherin (Fig. 2d). Indeed, volumes of the cavities in the P-cadherin apo-state structure (PDB ID; 4zmz), the E-cadherin structure (PDB ID; 2o72), and the P-cadherin monomer-Hit 1 complex structure calculated using CASTp[30] were 12.2 Å$^3$, 5.4 Å$^3$, 21.4 Å$^3$, respectively, suggesting that P-cadherin but not E-cadherin has a suitable cavity for the binding of Hit 1. Moreover, FTMap analysis[31] regarded the region as a binding cavity only in the P-cadherin structure, not in the E-cadherin structure (Supplementary Fig. S4). As expected, Hit 1 did not bind to E-cadherin but did bind to the E-cadherin N140Y mutant in the SPR-based direct binding assay (Fig. 2e). The N140Y point mutation did not affect the secondary structure of the recombinant E-cadherin, as confirmed using circular dichroism spectroscopy (Supplementary Fig. S5). Therefore, the binding cavity of Hit 1 could be used as a starting point for the design of a ligand selective for P-cadherin.

**Effect of Hit 1 on X dimerization.** To monitor the effect of Hit 1 on X dimerization, we used hydrogen–deuterium exchange mass spectrometry (HDX-MS). The method allowed us to monitor the extent to which the interface of X dimer is exposed to the solvent in the presence or in the absence of Hit 1. Pepsin treatment before LC-MS yielded peptide fragments from almost all regions of the P-cadherin molecule (Supplementary Fig. S6a, b, c). The HDX ratios for each peptide with and without Hit 1 were determined (Supplementary Fig. S6d, e). In the region that corresponds to one of the interfaces (residues 134–140 and 137–147) of MEC12[21], a construct that forms the X dimer but not the S–S dimer due to the inserted methionine at the N terminus of EC12 to inhibit the strand-swapping (see Supplementary Table S1 for a list of constructs), the HDX ratio in the presence of Hit 1 was higher than in the absence of Hit 1, although not as high as that of monomer, REC12 (Fig. 3a). This result suggests that the Hit 1 has some

Supplementary Table S1 for a list of constructs), was immobilized on the Sensor Chip SA via biotin–streptavidin capture method and 100 µM of each compound was injected onto the sensor chip surface. Based on the immobilization level and molecular weights of REC12 and fragment compounds, we estimated that the $R_{MAX}$ value of the binding response would be around 20 RU. Of the 1973 compounds in the library, 144 compounds had binding responses greater than 10 RU (Supplementary Fig. S1a).

As a secondary screen, we immobilized a construct called EC12 on the Sensor Chip CM5. This construct is in equilibrium between monomer and S–S dimer. We performed the so-called ABA assay, in which two solutions, A and B, are injected successively. In our experiments, 2 µM EC12 was used as solution A, and 100 µM of each of the compounds identified in the primary screen was used as solution B. In both A parts, the binding responses of monomer in the analyte and monomer covalently linked to the sensor chip surface were obtained. Two compounds clearly decreased the response in the B part (Supplementary Fig. S1b and Fig. 1c). These compounds bound to and disrupted the dimerization such that monomers that were not covalently immobilized on the sensor chip dissociated from the chip surface. Only one of the two compounds, Hit 1 hereafter, showed a dose–response dependency in the direct binding assay (Fig. 1d). The approximate $K_D$ value of Hit 1 (Fig. 1e) for the monomer mutant was calculated to be 916 µM.

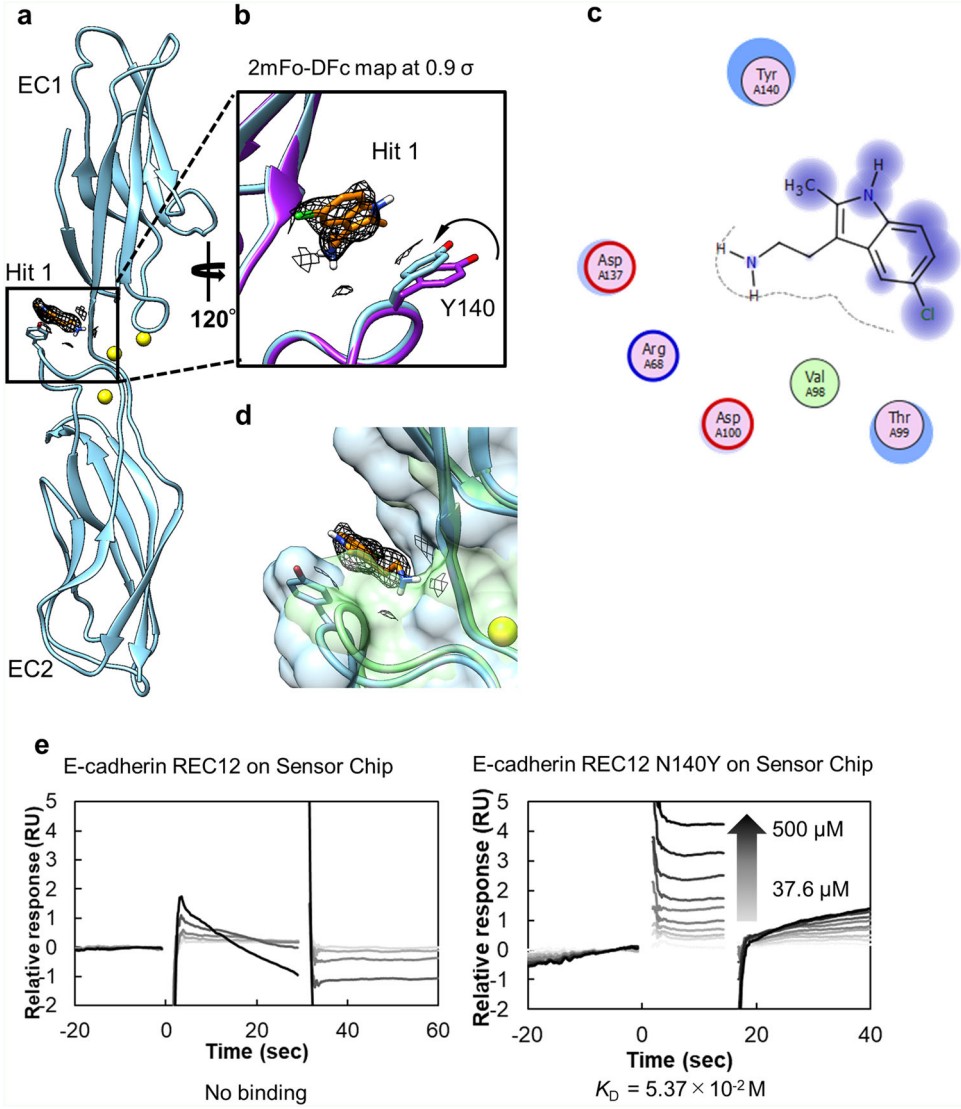

**Fig. 2 Hit 1 binds selectively to P-cadherin. a** Structure of complex of REC12 with Hit 1. Hit 1 is colored orange. **b** Region of the Hit 1-binding cavity, illustrating the shift in the side chain of Y140 that occurs upon compound binding. 2mFo-DFc map at 0.9 σ level is shown. The P-cadherin monomer (PDB ID; 4zmz) is superposed (purple). **c** 2D interaction map of Hit 1 made with FLEV[58]. The reduction of solvent exposure upon Hit 1 binding is indicated by the halo-like disc around the residues. **d** Superposition of P-cadherin REC12-Hit 1 complex and E-cadherin structure (PDB ID; 2o72, green). **e** Sensorgram from SPR-based binding assay of WT E-cadherin and E-cadherin N140Y with Hit 1. $N = 2$.

impact on the interface of X dimerization. The assay does not reflect all the residues at the interface between monomers in the X dimer (Supplementary Fig. S6f, g). This may be because the interface of the N terminal region involved in the interaction of the X dimer is too dynamic to analyze in HDX-MS, since the HDX reaction occurs in regions exposed to the solvent due to dynamics.

To further investigate how Hit 1 affects X dimerization, X dimer (MEC12) was crystallized and soaked with Hit 1 at 10 mM final concentration, resulting in the complex structure at 2.45 Å resolution (Fig. 3b and Table 1). Three independent sets of electron density that can be modeled as Hit 1 were found, two of them in the cavity around Y140, the same cavity in Fig. 2a, from each monomeric chain, and the other at the intersection of EC2 domains (Fig. 3b). The Hit 1 molecules bound in the cavity formed π–π interaction with Y140 and hydrogen bonds with residues or water molecules around the pocket (Supplementary Fig. S7). Since Hit 1 that binds to the cavity in X dimer forms more non-covalent interactions with the residues around the

pocket, the affinity of Hit 1 to X dimer seemed stronger than to monomer, which was confirmed by switchSENSE technology (Supplementary Fig. S8). The Hit 1 molecule at the intersection of EC2 domains bound mainly due to van der Waals interaction (Supplementary Fig. S7). The cavity around Y140 has a different structure through X dimerization from monomer and the binding mode of Hit 1 likely differs to fit the cavity.

A drastic structural change was observed in Hit 1-bound X dimer when compared with the apo-state of X dimer (PDB ID; 4zmq). The angle of EC1–EC2 domain became flatter and chain B was shifted relative to chain A (Fig. 3c). This structural change in angle can be explained most reasonably by two Hit 1 molecules bound in the cavity around Y140. Hydrogen bonds including one between the side chain of K14 and the main chain of A138' at the interface of X dimer were disrupted upon Hit 1 binding (Fig. 3d), which could explain the shifted location of two monomers.

To confirm that the hydrogen bond between the side chain of K14 and the main chain of A138' is important for X dimerization as previously reported[21], we used a size-exclusion chromatography-

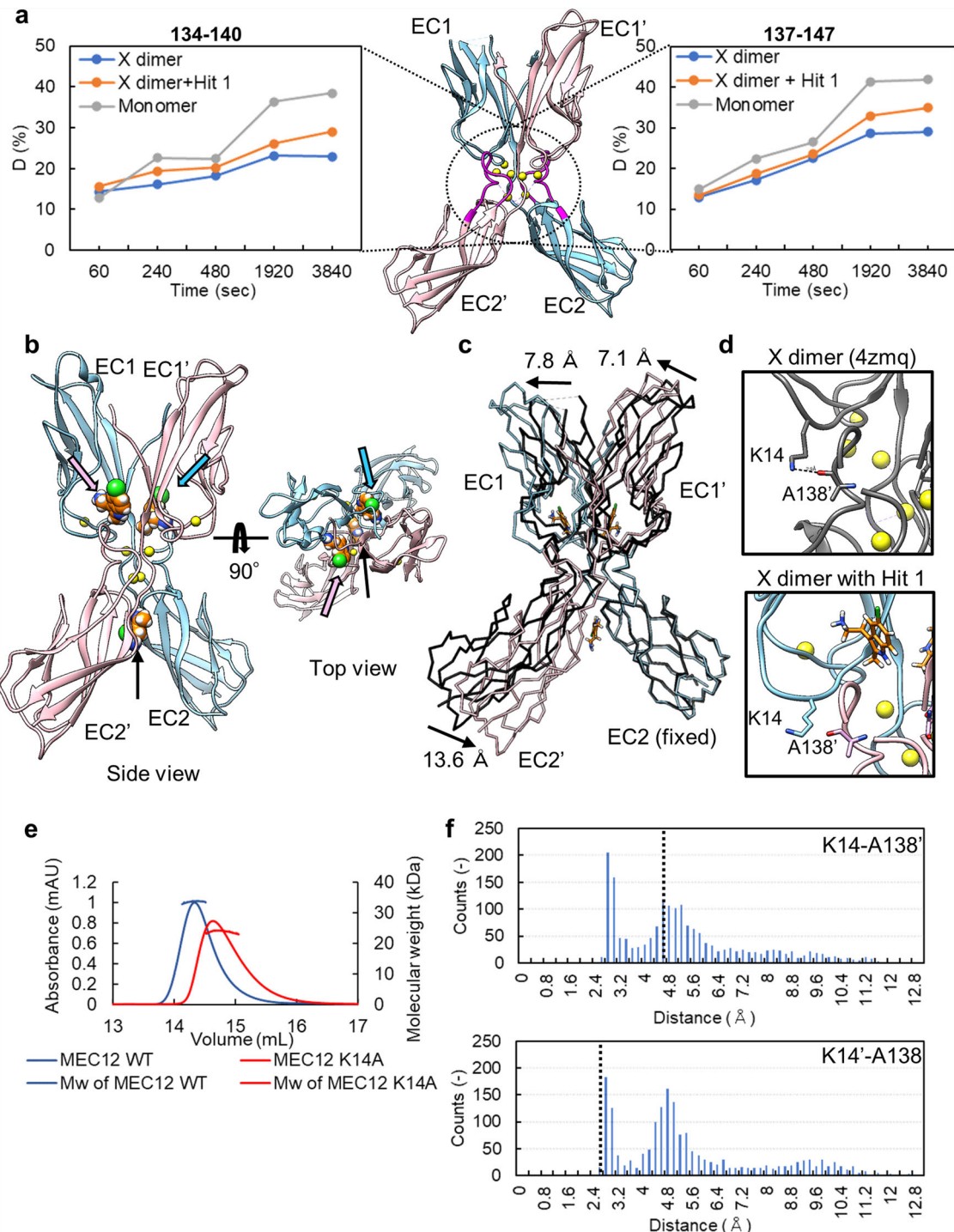

**Fig. 3 Effects on X dimerization by Hit 1. a** Hydrogen–deuterium exchange ratio of two peptides 134–140 and 137–147 as a function of time. The region of these peptides is colored in magenta in the structure of MEC12, X dimer (PDB ID; 4zmq). Chain A of X dimer is colored in sky blue; chain B in pink. **b** Structure of the complex of the X dimer and Hit 1. Hit 1 (in the sphere, orange) is bound in the cavity around Y140 of each monomeric chain and at the intersection of EC2 domains. The pink or sky blue or black arrows show Hit 1 that binds to chain A or chain B or at the intersection of EC2 domains, respectively. **c** Structural change caused by the binding of Hit 1. Hit 1-bound X dimer is aligned to the apo-state X dimer (PDB ID; 4zmq, black). The angle of EC1 to EC2 is flatter compared to that in the apo-state X dimer. The arrows indicate the movement of domains. **d** Hydrogen bonds disrupted by the structural change that occurs upon Hit 1 binding. In the apo-state X dimer, K14-A138′ hydrogen bond is shown in the dotted line. In Hit 1-bound X dimer, the K14-A138′ hydrogen bond is disrupted. Amino acid residues from chain A are shown in sky blue, those from chain B are in pink. **e** Size measurement using SEC-MALS. The WT X dimer trace is in blue, the K14A mutant trace is in red. $N = 1$. **f** Frequency histogram of the distance between a hydrogen bond donor and an acceptor of the pairs of K14-A138′ and K14′-A138 observed in the MD simulations of the apo-state X dimer (PDB ID; 4zmq). The black dotted lines represent the distances observed in the crystal structure of Hit 1-bound X dimer we obtained in this study.

multiangle light-scattering (SEC-MALS) analysis and measured the molecular weight of X dimer (MEC12) and X dimer (MEC12) K14A. Indeed, the K14A mutant was mainly in monomer form (Fig. 3e). The experimentally obtained molecular weight values are 33.8 kDa for X dimer and 24.2 kDa for X dimer K14A, while the calculated molecular weight from the polypeptide chain is 23.4 kDa as a monomer.

Since the key interaction between K14 and A138' was disrupted in the Hit 1-bound X dimer, we speculate that the structure of X dimer may represent one of the metastable states in the process of X dimerization. To prove the metastable states or solution-state dynamics of X dimer in the apo form, molecular dynamics (MD) simulation was performed using the apo-state structure of X dimer (PDB ID; 4zmq). Three independent simulations for 100 ns each were conducted and the convergence of the trajectories was confirmed by root mean square deviation (RMSD) of Cα atoms (Supplementary Fig. S9a). We then computed the distance of hydrogen bond donor (NZ) and acceptor (O) in the pairs of K14-A138' and K14'-A138, respectively (Fig. 3f), demonstrating that the frequency distribution has two peaks, the one at 2.6 Å and the other at 4.8 Å. This asymmetric nature was observed in all the trajectories (Supplementary Fig. S9b). In the crystal structure of the Hit 1-bound state X dimer, the distance of the pairs of K14-A138' and K14'-A138 were 4.6 and 2.6 Å, respectively. This result implies that apo-state dynamics of X dimer could have two possible states in terms of the hydrogen bond formation, and both states correspond to the ones observed in the crystal structure of the Hit 1-bound state of X dimer. In other words, the apo-state X dimer was able to assume conformations close to the ones Hit 1-bound X dimer exhibited in the crystal structure, even without Hit 1 compound. These observations suggest that the Hit 1-bound state of X dimer observed in the crystal structure we obtained is one state in the process of X dimerization. Collectively, Hit 1 binding may interfere with the formation of hydrogen bonds necessary for X dimerization by trapping X dimer into one state in the process of X dimerization. Unlike a typical orthosteric inhibitor, Hit 1 did not directly block hydrogen bonding between two monomer units; rather, Hit 1 may alter the monomer structure to prevent hydrogen bond formation.

**Inhibition of cell adhesion by Hit 1**. We next investigated whether or not Hit 1 could inhibit cell adhesion, using a previously established cell-aggregation assay[32,33]. We established a CHO cell line expressing P-cadherin using the Flp-In-CHO system. In this assay, extracellular proteins other than P-cadherin are trypsinized so that cell adhesion and formation of cell aggregates depend only on the interaction of P-cadherin molecules. Indeed, the mock CHO cells did not form large cell aggregates (Fig. 4a, c). After the trypsinization, the culture medium was replaced with a medium without calcium so that the cells were not aggregated. The aggregation reaction was initiated by the addition of 1 mM CaCl₂. After the cell-aggregation reaction, the size distribution of cell aggregates was measured using a microflow imaging (MFI) instrument. In MFI measurement, the number of cell aggregates >40 μm in size was counted as P-cadherin-dependent cell aggregates, since a considerable number of cell aggregates of the size 25–40 μm were observed in the presence of EDTA. In order to normalize the number of the counts, the ratio of the total number of particles larger than 40 μm to 100 μm over that of the particles >25–100 μm was calculated. In the control experiment, EDTA inhibited the formation of large cell aggregates (40–100 μm); thus adhesion is based on the calcium-dependent interaction of P-cadherin molecules (Fig. 4a, c). When Hit 1 and 1 mM CaCl₂ were added simultaneously, aggregation was inhibited in a manner that depended on the Hit 1 concentration

(Fig. 4b, c). When Hit 1 was added to pre-formed cell aggregates, the aggregates remained stable in the time scale of this assay; EDTA partially disrupted pre-formed aggregates (Supplementary Fig. S10). Considering these results, it may be that Hit 1 blocks association of two monomers in the process of the X dimerization. and that once cell aggregates are formed probably based on *cis* clustering, it is too stable to disrupt with Hit 1.

To quantify the effect of Hit 1 on the process of X dimerization, we performed a liposome aggregation assay. We first prepared C-terminally His-tagged X dimer (MEC12) and incubated the protein with a DOPC-based liposome that contained 10% Ni-chelating lipid DOGs-NTA-Ni in the presence of EDTA to deactivate P-cadherin molecules. Under these conditions, no aggregation was observed as monitored by absorbance at 650 nm (Fig. 4d). We also confirmed that liposome aggregation only happens in the presence of both MEC12 and CaCl₂ (Fig. 4d). Upon addition of CaCl₂ to the solution, liposomes aggregated, presumably through X dimerization of C-terminally His-tagged X dimer (MEC12) molecules captured on the surface of the liposomes as indicated by a gradual increase in absorbance at 650 nm that reached a plateau (Fig. 4d). The observed rate constant ($k_{obs}$) of the liposome aggregation was calculated as an indicator of the kinetics of X dimerization using an exponential decay equation model of the one-phase association. The $k_{obs}$ value was considerably lower in the presence of Hit 1 ($k_{obs} = 0.0033$) than in the absence of Hit 1 ($k_{obs} = 0.0086$) (Fig. 4e and Supplementary Fig. S11). Note that it was not the activation of cadherin molecules that Hit 1 inhibited, because calcium binding of cadherin molecules was not inhibited in the presence of Hit 1, which was shown by DSC measurement in the presence or absence of calcium or Hit 1 (Supplementary Fig. S12). This result supports our hypothesis that Hit 1 inhibits the process of X dimerization and presumably cell aggregation through the indirect disruption of hydrogen bonds necessary for X dimerization.

We also analyzed the effects of Hit 1 on cell adhesion mediated by endogenous P-cadherin and on cells that do not express P-cadherin using two cancer cell lines; HCT116 cells express P-cadherin[17], and MCF7 cells do not[34]. Cells were plated in the presence of 100 μM Hit 1 and the extent of cell adhesion was quantified indirectly by total cell area. Since the area from every single cell was different in HCT116 and MCF7, the normalized cell area defined as the total cell area in the presence of each concentration of Hit 1 over the total cell area in the absence of Hit 1 was calculated. The normalized cell area of HCT116 cells was decreased in the presence of 100 μM Hit 1 to 75.7% of the area in the absence of Hit 1 but Hit 1 had no effect on normalized MCF7 cell area (Fig. 4f). In the previous reports, the effect of P-cadherin on cancer metastasis is discussed based on the knockdown of P-cadherin and the resulting impact on signaling pathway[17,35]. This result, together with the phenomenon from the previous studies, indicates that Hit 1 blocks adhesion of cancer cells that express P-cadherin, which may result in the downregulation of the β-catenin pathway and activation of apoptosis of cells[17,35].

**Design of a more potential inhibitor**. Since Hit 1 and many of the compounds identified in the primary screen had cationic functional groups, we first checked whether the cationic functional group is important for binding activity by testing commercially available indole-based compounds: tryptamine, tryptophan, and auxin (Fig. 5a). Surprisingly, only tryptamine bound to P-cadherin (Supplementary Fig. S13a), which may be because the protein surface around the cavity where Hit 1 binds is negatively charged (Supplementary Fig. S13b). This result indicated that the cationic functional group should not be modified during further development.

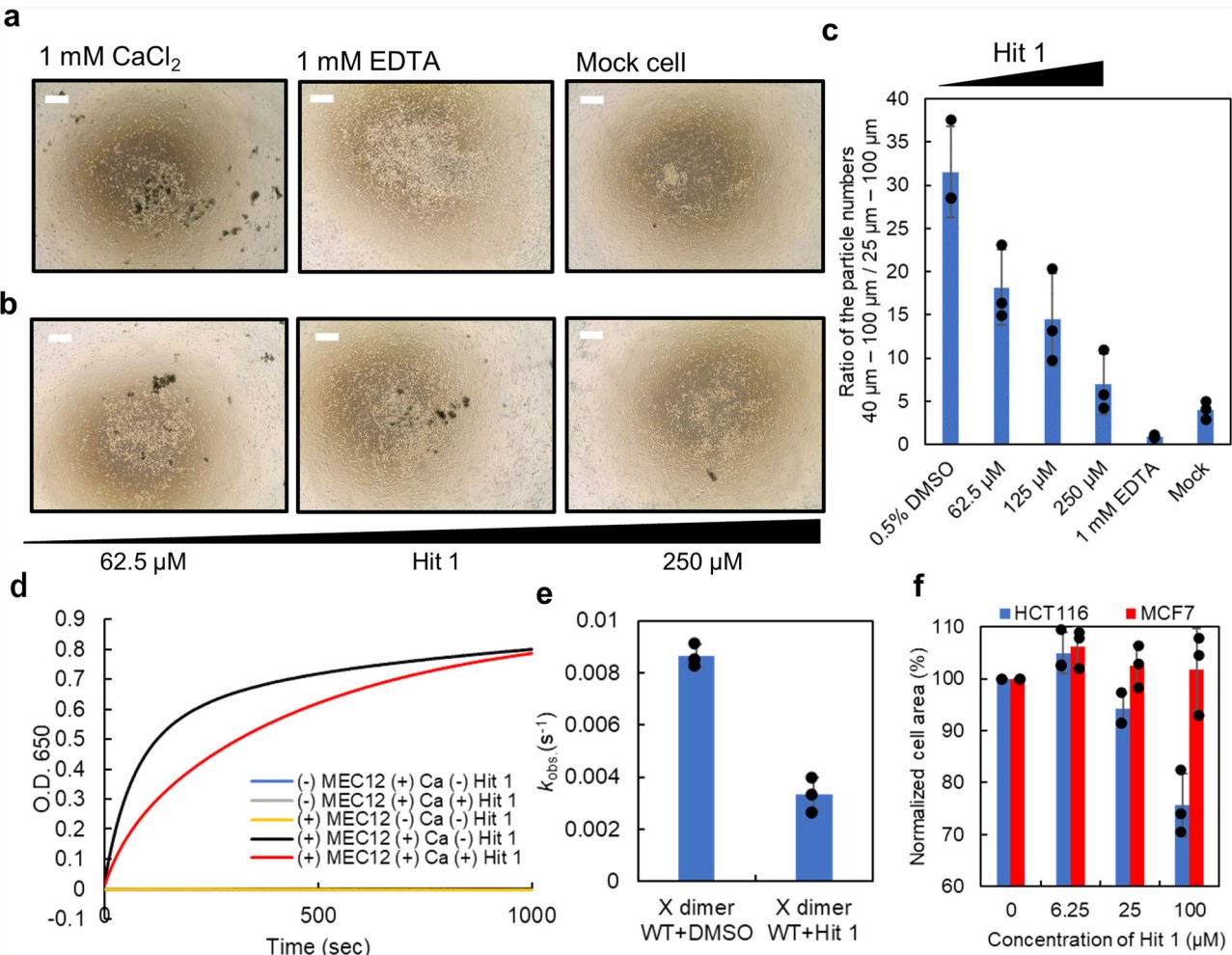

**Fig. 4 Hit 1 blocks aggregation of liposomes and cells with P-cadherin on their surfaces. a** Images of CHO cell line expressing P-cadherin in the presence of 1 mM CaCl$_2$ and 1 mM EDTA and a mock cell line. Scale bars indicate 500 μm. **b** Representative images of CHO cell line expressing P-cadherin in the presence of 1 mM CaCl$_2$ and increasing concentrations of Hit 1. This experiment was repeated three times, and similar results were obtained. Scale bars indicate 500 μm. **c** Quantification of cell-aggregation assay using MFI. Hit 1 inhibited the formation of cell aggregation in a dose-dependent manner. $N = 3$. Error bars show standard deviation. **d** Liposomes with MEC12 on the surface were incubated with CaCl$_2$ and with (red) or without (black) Hit 1. Controls had no effect on optical density (blue, yellow, and gray traces). A representative plot of optical density at 650 nm is shown as a function of time. $N = 3$. **e** The $k_{obs}$ of liposome aggregation was calculated assuming a one-phase association model. $N = 3$. Error bars show standard deviation. **f** Normalized area of HCT116 and MCF7 cells in the presence or absence of Hit 1. In order to take cell shape into account, the cell area in the absence of Hit 1 was taken as 100%. $N = 3$. Error bars show standard deviation. Individual data points are shown in black plots.

Upon study of the 2D interaction map of Hit 1 with the P-cadherin monomer (Fig. 2c), we came up with two strategies for the further synthesis: (1) replacement of the methyl group at the second position of the indole ring with a carboxy group, which should result in the formation of salt bridges between the carboxy group with R68 and the amino group with D137 and (2) replacement of the chlorine group at the C5 position of the indole ring with a bulkier functional group to inhibit approach of the second monomer. We first synthesized 3-(2-aminoethyl)-5-phenyl-1H-indole-2-carboxylic acid (Hit 1-carboxylic acid, Fig. 5b), but its affinity for REC12 in the SPR-based direct binding assay was no better than that of Hit 1 and it did not have inhibitory activity in cell-aggregation assay (Supplementary Fig. S14).

Next, we synthesized 2-(2-methyl-5-phenyl-1H-indol-3-yl) ethan-1-amine (phenyl-Hit 1) (Fig. 5b).

In the SPR-based direct binding assay, we observed a dose-dependent response; the binding affinity for REC12 was equivalent to that of Hit 1 (Supplementary Fig. S15). In the liposome aggregation assay, phenyl-Hit 1 had stronger inhibitory activity than Hit 1; tryptamine, auxin and tryptophan had little effect on aggregation (Fig. 5c and Supplementary Fig. S11). Tryptamine does not have a bulky functional group, which is likely why it did not inhibit liposome aggregation. In the cell-aggregation assay, phenyl-Hit 1 inhibited cell aggregation much more strongly than did Hit 1 (Fig. 5d), whereas negative controls and tryptamine did not (Supplementary Fig. S16). Given this strong inhibitory activity, the binding affinity of Phenyl-Hit 1 to X dimer could be stronger than to monomeric P-cadherin.

## Discussion

Through biophysical and structural methods, we identified a class of fragment compounds that bind to a unique shallow cavity on P-cadherin between the EC1 and EC2 domains. Our chemical fragment has the potential to modulate the process of X dimerization by altering the angle of the EC2 domain relative to the EC1 domain, blocking the formation of key hydrogen bonds

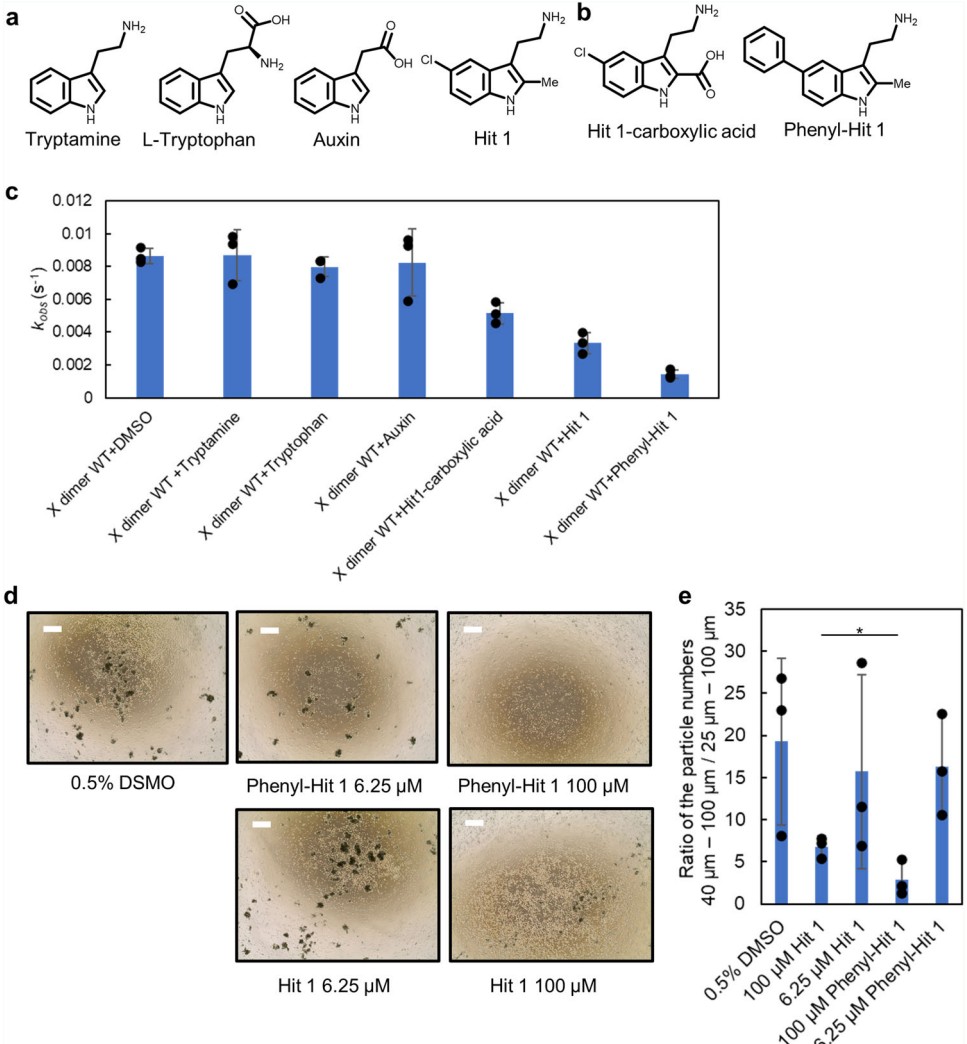

**Fig. 5 Phenyl-Hit 1 more effectively inhibits aggregation than does Hit 1. a, b** Chemical structures of the compounds investigated. **c** The $k_{obs}$ values for indicated compounds in the liposome aggregation assay. $N = 3$. Error bars show standard deviations. Individual data points are shown in black plots. **d** Representative images from cell-aggregation assay in the presence of a range of concentrations of phenyl-Hit 1. Scale bars indicate 500 μm. $N = 3$. In order to compare the inhibitory activity of Hit 1 and phenyl-Hit 1, the same lot of cells was used for both compounds. **e** Quantification of cell-aggregation assay. $N = 3$. Error bars show standard deviations. *Significant difference ($P < 0.05$ by one-sided Welch's $t$ test. $P = 0.032$).

necessary for X dimerization (Figs. 3c and 6). This is the first report of an inhibitor of a classical cadherin family protein that has a mechanism other than an orthosteric one. We propose that the restricted protein domain angle provides a common strategy to regulate PPI of cell-adhesive molecule with a small molecule since not only cadherin superfamily proteins but also integrin superfamily proteins and immunoglobulin superfamily proteins have multidomain structures and similar cavities located between the domains could be targeted to regulate the protein dynamics necessary for function.

The fragments identified herein had an impact on the dimerization process; probably by trapping X dimer to another state that lacks a key hydrogen bond. The SPR-based compound screen can provide an ideal platform to select such a dimerization modulator, since a flow system of SPR itself could help the inhibition of interaction, and thus the inhibitory mechanism should not be so strong as to be a thermodynamic one. A previous study of an anti-P-cadherin single-chain Fv (scFv) mentioned the importance of X dimerization in the regulation of cell adhesion[36]. Hit 1 modulated the process toward X dimerization, and despite the lack of a strong effect that can directly and

thermodynamically change the final equilibrium state of X dimer, it may have inhibited cell adhesion. This result implies that every stepwise process like the formation of cell adhesion assembly has a checkpoint, such as the intermediate like X dimer, that decreases the energy barrier to the final state and that a slight inhibition of the key step can lead to the inhibition of formation of the final state. In addition, the scFv, which had a nM range affinity to monomeric P-cadherin, was able to disrupt the pre-formed cell aggregates[37], whereas our chemical fragments were not. The cell-aggregation disruption assay may suggest that cell aggregates are too stable to regulate with a small molecule with a weak affinity. Therefore, the most effective small-molecule inhibitor of the formation of a stable macromolecule complex should block the initial association of the component proteins. In other words, it is this initial association of the component molecules that should be targeted for the drug discovery of the field.

From our simple structure–activity relationship study, we determined that an inhibitor with specificity for P-cadherin requires (1) a cationic functional group to interact with the negatively charged region around the cavity located between the EC1 and EC2 domains, (2) functional groups that interact with

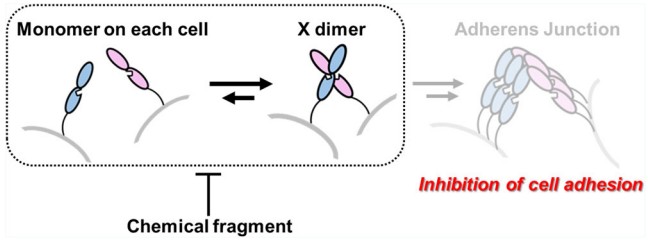

**Fig. 6 Inhibition of cell adhesion by a chemical fragment.** In the presence of chemical fragments, the angle between the EC1 and EC2 domains is not optimal for X dimerization; thus, the process of X dimerization was inhibited. The inhibition of the initial step in the whole process toward cell adhesion can be strong enough to inhibit cell adhesion.

Y140, a unique residue to P-cadherin, and (3) a bulky functional group that protrudes from the interface to block X dimerization. Our structure–activity relationship also suggests that strong affinity for the protein–protein interface itself is not strictly necessary for inhibition of protein complex formation since it is not the equilibrium but rather the first contact of component molecules that an inhibitor should affect.

We discovered that tryptamine but not tryptophan binds to P-cadherin. It is known that several tryptamine-derived metabolites have their own biological role[37,38]. One tryptamine derivative, serotonin, is important in the function of digestive organs including gastric tissue[39]; P-cadherin is overexpressed in gastric cancer. It is possible that tryptamine derivatives, through weak interactions with P-cadherin, subtly regulate cell adhesion. In support of this hypothesis, relationships between serotonin activity and cell adhesion phenotypes have been demonstrated[40–42].

In summary, the chemical fragment we identified acts to inhibit the formation of an intermediate in dimerization of P-cadherin. Cadherin family proteins, including P-cadherin, are associated with diseases such as cancer; thus, the small-molecule regulator of dimerization we identified has therapeutic potential. Further, our regulation strategy of protein–protein interactions and stepwise assembly of protein complexes using small molecules could be applied to identify inhibitors of the formation of other macromolecular complexes.

## Methods

**Protein expression and purification.** For the in vitro assays including SPR-based screen, crystallography, HDX-MS, and SPR-based selectivity analysis, human P-cadherin constructs and human E-cadherin constructs were expressed in *E. coli* Rosetta2(DE3). *E. coli* cells were transformed with the pET SUMO vector and used to inoculate 6 mL LB medium containing 50 μg/mL kanamycin and 34 μg/mL chloramphenicol. Cells were pre-cultured at 37 °C for 16 h then transferred into 1 L of fresh LB medium containing the same antibiotics and cultured again at 37 °C for 4 h. At this time, the O.D.$_{600}$ was around 0.5. Isopropyl β-D-1-thiogalactopyranoside (IPTG) was added to 0.5 mM to induce recombinant protein expression. After 16 h at 20 °C, *E. coli* cells were collected by centrifugation at $7000 \times g$ at 4 °C for 10 min, suspended in binding buffer (20 mM Tris, 300 mM NaCl, 3 mM CaCl$_2$, 20 mM imidazole, pH 8.0), and sonicated for 10 min. The lysate was centrifuged at $40,000 \times g$, 4 °C, for 30 min. The supernatant was purified on a Ni-NTA agarose (QIAGEN) column, pre-equilibrated with the binding buffer. The His-tagged protein was eluted with elution buffer (20 mM Tris, 300 mM NaCl, 3 mM CaCl$_2$, 300 mM imidazole, pH 8.0), and incubated in SEC buffer (10 mM HEPES, 150 mM NaCl, 3 mM CaCl$_2$, pH 7.5) with Ulp1 protease to remove the SUMO protein. The resultant protein was again loaded onto a Ni-NTA agarose column and the flow-through was further purified with size-exclusion chromatography (SEC). The protein was loaded onto a Hiload 26/60 Superdex-200 column (Cytiva) pre-equilibrated with SEC buffer. All cadherin constructs were based on EC12 (1–241), which consists of two extracellular domains. For the control experiments in the SPR-based direct binding screen, anti-P-cadherin scFv TSP7 was prepared. The expression host was *E. coli* BL21 (DE3). The binding buffer contained 20 mM Tris, 500 mM NaCl, 5 mM imidazole, pH 8.0. The elution buffer was 20 mM Tris, 500 mM NaCl, 300 mM imidazole, pH 8.0. In SEC process, a Hiload 26/60 Superdex-75 column (Cytiva) was used in 20 mM Tris, 200 mM NaCl, pH 8.0 buffer.

**Compounds.** The fragment library used in the SPR-based screen was purchased from Drug Discovery Initiative. Hit 1 was purchased from Vitas-M Laboratory. For HDX-MS experiments, SPR-based selectivity analysis, and the mutation study, Hit 1 was synthesized in-house. The synthetic scheme, procedures, and compound characterization are described in Supplementary Methods. Tryptamine, auxin, and L-tryptophan were purchased from Sigma Aldrich or Nacalai Tesque. The stock solution of all the compounds was prepared in DMSO and stored at −30 °C.

**Direct binding analysis using SPR.** For the primary screen, we performed a direct binding assay using SPR. All the SPR-related experiments were performed on a Biacore 8 K (Cytiva) at 25 °C. The monomer construct of REC12 was immobilized on the Sensor Chip SA via biotin–streptavidin capture. The immobilization level was ~3000 RU. Fragment compounds were injected onto the sensor chip surface in running buffer (10 mM HEPES, 150 mM NaCl, 3 mM CaCl$_2$, 0.05% Tween20, 5% DMSO, pH 7.5) to a final concentration of 100 μM. Both association time and dissociation time were 20 s, and 1 M arginine–HCl, pH 4.4 was used to regenerate the sensor chip surface. The running buffer alone was used as a negative control, and 1 μM TSP7 was used as a positive control. The solvent correction was performed periodically. The binding response of each compound was normalized to those of control samples and molecular weight using the Biacore 8 K software.

**ABA assay using SPR.** For the secondary screen, we performed an ABA assay. EC12 was immobilized on the Sensor Chip CM5 by amine coupling (pH 4.5). The immobilization level was approximately 500 RU. Solution A was 2 μM EC12, and 100 μM compound identified in the primary screen was solution B. The association time for the first injection of solution A part was 180 s, that of solution B was 120 s, and that of the second injection of solution A was 120 s.

**Selectivity analyses and mutation study.** E-cadherin WT REC12, E-cadherin N140Y mutant, P-cadherin WT REC12, and P-cadherin Y140N mutant were immobilized on chips via biotin–streptavidin capture. All the constructs gave the binding response between 3000–4000 RU. Association times were 15 or 30 s, and dissociation time was 20 s. Hit 1 concentration ranged from 37.6 to 600 μM. The $K_D$ values were calculated using the Scatchard method using the Biacore 8 K software. The solvent correction was performed at the beginning and the end of the concentration series.

**Circular dichroism (CD) measurement.** To confirm that point mutations of classical cadherin constructs did not disrupt the secondary structure, we performed CD measurements. CD spectra of samples in 1 mm path-length quartz cells were measured at 20 °C using a JASCO J-820 spectropolarimeter. Samples were prepared at 10 μM in 10 mM HEPES, 150 mM NaCl, 3 mM CaCl$_2$, pH 7.5 buffer.

**Crystallization of P-cadherin.** For X-ray crystallography, conditions used previously for the C-terminal-deleted REC12 and MEC12 (1–213)[21] were used. Purified C-terminal-deleted REC12 (12.5 mg/mL) was crystallized in 100 mM HEPES, pH 7.5, 28% v/v PEG 400, 200 mM CaCl$_2$. Purified C-terminally-deleted MEC12 (12.5 mg/mL) was crystallized in 0.17 M sodium acetate trihydrate, 0.085 M Tris pH 8.5, w/v 25.5% PEG4000, 15% v/v glycerol. The crystals were soaked with the crystallization solution containing 10% DMSO and 10 mM Hit 1 for several minutes. For the crystals of C-terminal-deleted REC12, crystal annealing[43] was performed in order to improve mosaicity.

**Data collection and refinement.** X-ray diffraction data sets were collected on the RIKEN Structural Genomics Beamline II (BL26B2) at SPring-8[44] at 100 K and a wavelength of 1.0 Å. The diffraction data were processed with the KAMO[45–47], and the structure was solved by molecular replacement, using the structure of P-cadherin REC12 (PDB ID 4zmz) or MEC12 (PDB ID 4zmq) as a search model with phenix.phaser. The resultant structures were iteratively refined using phenix.refine[48] and manually rebuilt in Coot[48]. Final Ramachandran statistics are as follows; 94.76% favored and 0.48% outliers in REC12-Hit 1 complex, 96.46% favored, and 0.47% outliers in MEC12-Hit 1 complex. The other final refinement statistics are summarized in Table 1. Figures were prepared with UCSF Chimera[49].

**HDX-MS homodimer-inhibition assay.** To confirm the hypothesis from the crystallography that Hit 1 inhibits X dimerization, we performed HDX-MS and monitored the protein surface exposed to the solvent with or without Hit 1. We compared the extent to which the X dimer interface is exposed to the solvent by preparing REC12 and MEC12 at 1.5 mg/mL protein in the H$_2$O-based 10 mM HEPES, 150 mM NaCl, 3 mM CaCl$_2$, pH 7.5, 5% DMSO with or without 2 mM Hit 1. The hydrogen–deuterium exchange reaction was started by diluting D$_2$O-based buffer by tenfold and was quenched by the addition of an equal volume of pre-chilled quenching buffer (8 M urea, 1 M Tris(2-carboxyethyl)phosphine hydrochloride, pH 3.0) with the HDx-3 PAL (LEAP Technologies). The quenched protein samples were subjected to online pepsin digestion and analyzed by LC-MS using UltiMate3000RSLCnano (Thermo Fisher Scientific) connected to the Q Exactive HF-X mass spectrometer (Thermo Fisher Scientific). Online pepsin digestion was performed using a Poroszyme Immobilized Pepsin Cartridge

(2.1 × 30 mm; Thermo Fisher Scientific) in formic acid solution, pH 2.5 at 8 °C for 3 min at a flow rate of 50 μL/min. The desalting column and the analytical columns were Acclaim PepMap300 C18 (1.0 × 15 mm; Thermo Fisher Scientific) and Hypersil Gold (1.0 × 50 mm; Thermo Fisher Scientific), respectively. The mobile phases were 0.1% formic acid solution (A buffer) and 0.1% formic acid containing 90% acetonitrile (B buffer). The deuterated peptides were eluted at a flow rate of 45 μL/min with a gradient of 10–90% of B buffer in 9 min. Mass spectrometer conditions were as follows: an electrospray voltage of 3.8 kV, positive ion mode, sheath and auxiliary nitrogen flow rate at 20 and 2 arbitrary units, ion transfer tube temperature at 275 °C, auxiliary gas heater temperature at 100 °C, and a mass range of $m/z$ 200–2000. The data-dependent acquisition was performed with normalized collision energy of 27 arbitrary units. The MS and MS/MS spectra were subjected to a database search analysis using the Proteome Discoverer 2.2 (Thermo Fisher Scientific) against an in-house database containing the amino acid sequence of the C-terminal-depleted REC12. The search results and MS raw files were used for the analysis of the deuteration levels of the peptide fragments using the HDExaminer software (Sierra Analytics).

**Kinetics measurement using switchSENSE**. Kinetics parameters upon the binding of Hit 1 onto MEC12 (X dimer) or REC12 (monomer) were measured on a heliX+ instrument (Dynamic Biosensors GmbH, Martinsried, DE) using the switchSENSE technology[50] in static fluorescence proximity sensing mode. Biotinylated MEC12 or REC12 construct was immobilized on the ADP-48-2-0 Biochip with the Biotin Capture Kit (HK-SA-1), using the prehybridized Adapter-1-Ra strand with streptavidin and prehybridized Adapter-2-Ra ligand-free strand as a reference. Hit 1 was prepared in 10 mM HEPES, 150 mM NaCl, 3 mM CaCl$_2$, 0.05% Tween20, 5% DMSO, pH 7.5 buffer in dilutions from 2 mM to 31.3 μM, and injected onto the sensor chip at 200 μL/min at 23 °C. The consecutive dissociation was done at 200 μL/min at 23 °C for 60 s. Kinetics parameters were calculated using the proprietary heliOS software with a buffer and real-time referencing.

**MD simulation**. Molecular dynamics simulations were performed using GROMACS 2016.3[51] with the CHARMM36m force field[52]. A crystal structure of apo-state X dimer (PDB ID: 4zmq) was used as the initial coordinates. The protein was solvated with TIP3P water[53] in a rectangular box such that the minimum distance to the edge of the box was 15 Å under periodic boundary conditions through the CHARMM-GUI[54]. Na$^+$ or Cl$^-$ ions were added to imitate a salt solution of concentration 0.15 M. Each system was energy-minimized for 5000 steps and equilibrated with the NVT ensemble (303 K) for 1 ns. Further simulations were performed with the NPT ensemble at 303 K. The time step was set to 2 fs throughout the simulations. A cutoff distance of 12 Å was used for Coulomb and van der Waals interactions. Long-range electrostatic interactions were evaluated using the particle mesh Ewald method[55]. Covalent bonds involving hydrogen atoms were constrained by the LINCS algorithm[56]. A snapshot was saved every 10 ps. All trajectories were analyzed using GROMACS with the converged trajectories from 40 ns to 100 ns (Supplementary Fig. S9a).

**SEC-MALS analyses**. In order to measure the molecular size of each mutant, SEC-MALS analysis was performed. Purified mutants were concentrated to 3.2 mg/mL in SEC buffer and loaded onto a 10/300 Superdex-200 column (Cytiva). Size measurement was performed using a Heleos 8+ instrument (Wyatt Technology) equipped with a triple MALS/refraction index (RI)/ultraviolet detector.

**Preparation of CHO cells expressing P-cadherin**. To evaluate cell adhesion, we used the Flp-In-CHO system (Life Technologies) to engineer CHO cells stably expressing full-length P-cadherin. A single clone was obtained through the limiting dilution-culture method. The expression of P-cadherin was monitored with an imaging cytometer (In Cell Analyzer 2000, Cytiva), because the DNA sequence of monomeric GFP was fused at the C-terminal of the human P-cadherin constructs. CHO cells expressing P-cadherin were cultured in Ham's 12 medium (Life Technologies) containing 10% fetal bovine serum (FBS), 1% penicillin–streptomycin, and 0.5 mg/mL hygromycin at 37 °C, 5% CO$_2$.

**Cell-aggregation assay**. The cell-aggregation assay was performed as reported previously[32,33]. We first treated the CHO cells with 0.1% trypsin in HEPES-based magnesium-free buffer (HMF buffer, 10 mM HEPES, 137 mM NaCl, 5.4 mM KCl, 0.34 mM Na$_2$HPO$_4$, 1 mM CaCl$_2$, 5.5 mM glucose, pH 7.4). These conditions result in the digestion of protein molecules on the cell surface with the exception of P-cadherin, which is resistant to trypsinization in the presence of Ca$^{2+}$. The trypsinization was stopped by adding HMF buffer containing 10% FBS, and trypsin was removed by washes with HMF buffer. After washing the cells with Ca$^{2+}$-depleted HMF buffer (HCMF buffer), cells were disaggregated. In total, 500 μL of cell solution containing 1 × 10$^5$ cells/mL was placed in wells of a 24-well plate that had been pre-treated with 1% (w/v) BSA. Addition of 1 mM CaCl$_2$ in 2% DMSO initiated P-cadherin-mediated aggregation reaction. Simultaneously with CaCl$_2$ addition, 1 mM EDTA in 0.5% DMSO as control or Hit 1 was added. Cells were incubated at 37 °C, 80 rpm for 30 min, and images of the cells were taken with an EVOS XL Core Imaging System (Life Technologies). To determine Hit 1 could

disrupt the cell aggregation, Hit 1 or EDTA was added 60 min after the addition of CaCl$_2$. The images of the cells were taken after 60 min.

**Microflow imaging (MFI)**. Micro-Flow Imaging (Brightwell Technologies) was used in order to measure the size distribution of cell aggregates. After the cell-aggregation assay, the plate was incubated at room temperature for 10 min and 125 μL of chilled 4% paraformaldehyde phosphate buffer solution (Nacalai Tesque) was added to each well. The plate was incubated at r.t. for more than 1 h. After that, cells were injected into MFI. After the MFI measurement, the number of cell aggregates larger than 40 μm in size was counted as P-cadherin-dependent cell aggregates.

**Liposome aggregation assay**. Lipids (DOPC: DOGs-NTA-Ni, 9:1 molar ratio) were dissolved in chloroform than dried. The lipids monolayer obtained was hydrated with 10 mM HEPES, 150 mM NaCl, 3 mM CaCl$_2$, pH 7.5 and subjected to ten cycles of freeze and thaw. Liposomes were prepared using a polycarbonate filter with 100-nm pore diameter in a Mini-Extruder apparatus (Avanti).

Based on a reported methodology[57], liposomes with C-terminally His-tagged MEC12 were prepared at a lipid to protein molar ratio of 50:1 in 10 mM HEPES, 150 mM NaCl, 3 mM CaCl$_2$, pH 7.5. After 10 min at room temperature, an excess amount of EDTA (8.3 mM) was added to deactivate the cadherin molecules. After the addition of 10 mM CaCl$_2$, the optical density at 650 nm was measured using a spectrophotometer every 1 s for 1000 s. Hit 1 or phenyl-Hit 1 diluted in DMSO was added to the solution to a final DMSO concentration of 5% DMSO. The constant rate $K$ was calculated using GraphPad Prism 8 software assuming exponential decay and one-phase association.

**Differential scanning calorimetry (DSC)**. DSC measurement was performed using an PEAQ-DSC instrument (Malvern). Samples were prepared at 1.0 mg/mL in 10 mM HEPES (pH 7.5), 150 mM NaCl with or without 3 mM CaCl$_2$, 5% DMSO, 1 mM Hit 1. Samples were heated from 20 to 110 °C at a rate of 60 °C/h.

**Cell area quantification**. HCT116 (National Institutes of Biomedical Innovation, Health and Nutrition, Japan) and MCF7 cells (National Institutes of Biomedical Innovation, Health and Nutrition, Japan) were detached from the plate by trypsin and resuspended to the concentration of 1 × 10$^5$ cells/mL in McCoy's 5 medium containing 10% FBS, and 1% penicillin–streptomycin, 0.1% DMSO, and each concentration of Hit 1. Aliquots of 100 μL of the suspended cells were plated into wells of a 96-well plate (Greiner) and incubated at 37 °C, 5% CO$_2$ for 2 days. After removing the medium, 100 μL of 10 μM Calcein AM (Invitrogen) in McCoy's 5 medium was added to each well and incubated at 37 °C, 5% CO$_2$ for 30 min. After washing with PBS buffer, 100 μL of PBS buffer was added, and images were taken using an In Cell Analyzer 2000 (Cytiva). The cell area was calculated using a method created with Developer Tool Box software. The kernel size and intensity were adjusted for each cell to fully detect the area.

**Statistics and reproducibility**. The number of replicates for each experiment was mentioned in the figure legends. Cell-aggregation assay and cell area quantification were performed using the same sample to make three replicates. The other experiments were performed using independently prepared samples. All the error bars show means ± SD. Welch's $t$ test was used for comparison of cell-aggregation samples in the presence of Hit 1 or phenyl-Hit 1 (*$P < 0.05$).

**Reporting summary**. Further information on research design is available in the Nature Research Reporting Summary linked to this article.

## Data availability
Coordinates and structure factors for the structures are deposited in the Protein Data Bank under accession codes 7CME and 7CMF. The authors declare that the data supporting the findings of this study are available within the paper and its supplementary information files. The source data used to make charts in the main figures are given as Supplementary Data 1.

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

## Acknowledgements

We thank Thermo Fisher Scientific for the technical support in HDX-MS experiments. We thank Dynamic Biosensors for the technical support in heliX measurement. The super-computing resource was provided by Human Genome Center, the Institute of Medical Science, the University of Tokyo. This work was supported by JSPS Grant-in-Aid for Scientific Research on Innovative Areas "Bio-metal" (19H05766 to K.T.), JSPS Grants-in-Aid for Scientific Research (16H02420 to K.T.), Platform Project for Supporting Drug Discovery and Life Science Research (Basis for Supporting Innovative Drug Discovery and Life Science Research (BINDS)) from Japan Agency for Medical Research

and Development (AMED) (JP19am0101094 to K.T.), and a Grant-in-Aid for JSPS fellows (A.S.). The synchrotron radiation experiments were performed at BL26B2 of SPring-8 and were supported by BINDS from AMED (JP19am0101070). The fragment compounds were provided by Drug Discovery Initiative supported by BINDS from AMED (JP19am0101086). This work was partly supported by the World-leading Innovative Graduate Study Program for Life Science and Technology, The University of Tokyo, as part of the WISE Program (Doctoral Program for World-leading Innovative & Smart Education), MEXT, Japan.

## Author contributions

A.S., S.N., S.K., T.T. and K.T. designed experiments, analyzed and discussed the results, and approved the manuscript. A.S., K.Y., S.I. and G.U. performed crystallization experiments, and processed and determined the crystal structure. A.S. and Y.S. synthesized Hit 1, Hit 1-carboxylic acid and phenyl-Hit 1 with input from S.S. A.S. performed MD simulations with input from D.K. A.S., D.K. and S.N. wrote the manuscript with input from K.T.

## Competing interests

The authors declare no competing interests.
