## [Peer Review File · Communications Biology]

Reviewers' comments:

Reviewer #1 (Remarks to the Author):

This is an interesting manuscript providing novel information and approaches to our understanding of cell-cell junctions.

The one experiment I feel is missing is the SEC-MALS of the protein-hit 1 complex.

The Figures and crystallography can be improved.

Supplementary Table 1 (please move into the main text)

- The Rmerge and Rmeas seem odd for the last shell
- The signal to noise too low in the last shell
- The number of solvent molecules seem too low at the given resolutions
- Please provide MolProbity scores

Supplementary Figure S2 and S7a-c – please provide all surrounding protein atoms

Supplementary Figure S4 – please enlarge the Figure to fill the page and provide sufficient labels

Supplementary Figure S6a through S6c – please enlarge to fill the entire page so that one can read it and replace the yellow with black

Supplementary Figure S6g – please show the monomer without surface in black, enlarge Figure and provide enough labels

Supplementary Figure S7d-f, 2c – please remove colors and increase font

Supplementary Figure S8, S11b, 4a, 4b – please increase brightness (S13 and 5d perhaps also)

Supplementary Figure S9 – please show all repeats and the raw data, not just the fitted curves

Supplementary Figure S10a – please enlarge and increase font

Supplementary Figure S10b – please enlarge and provide labels

Supplementary Figure S11a, S12, 1d, 2e – please use a different color for each trace and label

Figure 2a, 2b, 2d, 3a-3d – please remove surface and provide sufficient labeling

Figure 3e – please replace the light blue with a bright red trace, please add molecular weights as calculated from polypeptide chain and as obtained from the experiment

Figure 4c – please show data and not just fitted curve for each repeat

Figure 4d and 4e – please provide the actual values also

Reviewer #2 (Remarks to the Author):

Summary

This paper reports a chemical fragment that disrupts dimerization of P-cadherin (cadherin 3). The

authors use a screen for cadherin-binding drug-like fragments and show using structural and biophysical techniques that the top hit compound binds to P-cadherin and lowers the on-rate of dimer formation. They then use basic cell biology experiments to suggest that the novel hit fragment inhibits cadherin-mediated cell aggregation/adhesion. The paper describes a valuable small-molecule screen, which yielded a potentially useful tool fragment candidate, with important potential applications that should be of interest to the field of adhesion biology. However, additional cell biological data are required to strengthen the conclusions and relevance to cell adhesion.

Major points

1. The authors should better introduce the hypothesis for targeting the hydrophobic pocket by expanding their argument in the Introduction with a more detailed rationale (p. 4, lines 56–59). It would be helpful to define more precisely for the reader why S–S dimer inhibition may not be the most appropriate approach, as alluded to in the manuscript.
2. The explanation of the structural change in Hit 1-bound X dimer (p. 13, lines 168–175) is not clear to me and would benefit from further corroboration. What is the evidence that two Hit 1 molecules bind in the cavity (or do the authors mean one Hit 1 molecule in each of the respective chains of the dimer)? What is the evidence that the Hit 1-bound dimer is in a metastable state? More data or more detailed explanations are required to support these conclusions.
3. The authors should provide evidence that only P-cadherin remains on the cell surface following trypsinisation, as claimed in the manuscript, as this is central to the principle of the P-cadherin-dependent cell aggregation assay.
4. Can the authors quantify the cell aggregation data from the multiple experimental replicates (Figs 4 and 5) to provide firmer cell biological conclusions?
5. Findings from the cell aggregation assay should be verified using an alternative, quantitative assay (e.g. quantify cell–cell aggregation by calculating particle size distribution after incubation of cells in suspension).
6. It is not clear how cell area was determined in Fig. 4e – is this mean area of an average cell or is it the proportion of the plate covered by cells (i.e. confluency)? More methodological details are required to interpret this experiment. Furthermore, the biological relevance of this assay is not clear: the association between cell area and cell–cell adhesion, in particular the relevance to P-cadherin-dependent cell adhesion, needs to be explained.
7. Can the authors explain why an order of magnitude higher concentration of Hit 1 is required to inhibit cell aggregation as compared to cell area?
8. The dosing range of phenyl-Hit 1 does not appear appropriate for the cell aggregation assay in Fig. 5d. A low dose at which aggregation is not inhibited should be used – the small aggregates at 62.5 μM indicated by an arrow are insufficient to convincingly show that the cells were healthy and able to aggregate normally in this experiment. In addition, this experiment should be performed alongside a positive and a negative control, and also alongside the original Hit 1 fragment to enable accurate comparison.

Minor points

1. Please use exact n number for number of independent experiments tested in Figs 1d, 2e, 3e, 4a, 4b, 4c and 5d.
2. The different construct names (e.g. REC12, MEC12) are not intuitive so should be summarised in a figure panel or table.
3. The fourth and fifth sentences of the Introduction (p. 3, lines 34–41) were difficult to follow and their clarity should be improved.
4. Make it clearer in Fig. 1 that the arrowed numbers in parentheses (e.g. 2 \rightarrow 1) represent decreases in the number of fragment candidates.
5. The angstrom symbol was not reproduced in the manuscript.
6. The label font in Fig. 2c is too small to read.

Reviewer #3 (Remarks to the Author):

In this study, the authors describe the identification and characterization of a compound that binds a region between P-cadherin domains EC1 and EC2 and blocks P-cadherin homo dimerization. They use several different approaches to characterize Hit 1 binding and inhibition of homo dimerization and cell adhesion, including mutagenesis, SPR, crystallography, mass spec, and light scattering. They speculate, based largely on the position of the binding site of Hit 1 that this compound acts by impeding the kinetics of homo dimerization.

Although the results are interesting and provide one mechanism for inhibiting cadherin binding, the argument regarding the mechanism of action is speculation because they don't actually show that the compound alters the binding kinetics, rather than the homo dimerization affinity. Some of the experiments also need additional controls. Detailed comments are given below.

1. All SPR measurements were done under equilibrium binding conditions and do not report the effect of Hit1 on P-cadherin dimerization kinetics or binding affinity. An alternative interpretation is that Hit1 allosterically alters the dimerization affinity. The authors need to quantify the influence of Hit1 on the cadherin dimerization affinity to rule this out and to support their assertion that Hit1 affects kinetics. SPR measurements of the dimerization kinetics +/- Hit1 are also needed to support their arguments.

For example, compare P-cadherin dimerization kinetics by immobilizing REC12 with Hit1 or low calcium to disrupt homo dimers, and then compare the rate and limiting amount of homo dimerization in the presence of Hit1.

2. Based on structure analysis, they mutated the Hit1 binding site, but did not test whether the Y140R mutation affects P-cadherin dimerization. This would support their claim on lines 116-117 that disrupting H-bonding around the binding site affects P-cadherin binding.

Secondary structure measurements do not address this because even calcium depletion does not completely disrupt the secondary structure, even though it blocks dimerization.

3. In liposome aggregation measurements to demonstrate kinetic effects, the authors add calcium and Hit1 simultaneously. They attribute reduced aggregation kinetics to altered X-dimer kinetics. They do not actually show this. As performed in this study, liposome aggregation convolves protein refolding/activation and cadherin binding. An alternative explanation is that Hit1 impedes calcium binding and cadherin activation. They should show altered kinetics at the protein level

4. On line 148-149, what is MEC12? Clearly state what this protein is in the text.

5. Measure Hit1 binding to MEC12 and compare with REC12. Are the affinities different?

6. Line 174, what do they mean by 'metastable state'?

7. Based on the evidence, argument in lines 180-183 is speculation only.

8. The inability of Hit1 to disrupt cell aggregates does not prove the equilibrium is unaffected. This could be due to diffusion limitation in dense cell aggregates. The authors should show this at the protein level with SPR.

9. In most of their figures, lettering on graphs is too small (especially SPR data) and is barely readable.

10. Technical details: How much P-cadherin dimer is bound initially? If they immobilize in REC12 in low calcium solution or the presence of Hit1, does this change the amount of cadherin initially loaded on the chip? How does this affect their Hit1 affinity estimations?

Response to the referees' comments and revisions that have been made

We thank the editor and the reviewers for the constructive comments on our manuscript. We revised the manuscript in light of the all comments. Please check the following point-by-point response to the reviewers' comments as well as attached the revised files ("Revised_manuscript.pdf" and "Revised_SI.docx"). We also attached copies of the revised files showing track changes ("Revised_manuscript_track_change.docx" and "Revised_SI_track_change.docx") for the benefit of the reviewers. We hope these answers would resolve all the reviewers' concerns.

(black: comments from the reviewers, blue: our comments, red: inserted/added/modified sentences/figures/tables)

Reviewer 1

Original comments from Reviewer 1

This is an interesting manuscript providing novel information and approaches to our understanding of cell-cell junctions.

The one experiment I feel is missing is the SEC-MALS of the protein-hit 1 complex.

The Figures and crystallography can be improved.

Supplementary Table 1 (please move into the main text)

- The Rmerge and Rmeas seem odd for the last shell
- The signal to noise too low in the last shell
- The number of solvent molecules seem too low at the given resolutions
- Please provide MolProbity scores

Supplementary Figure S2 and S7a-c – please provide all surrounding protein atoms

Supplementary Figure S4 – please enlarge the Figure to fill the page and provide sufficient labels

Supplementary Figure S6a through S6c – please enlarge to fill the entire page so that one can read it and replace the yellow with black

Supplementary Figure S6g – please show the monomer without surface in black, enlarge Figure and provide enough labels

Supplementary Figure S7d-f, 2c – please remove colors and increase font

Supplementary Figure S8, S11b, 4a, 4b – please increase brightness (S13 and 5d perhaps also)

Supplementary Figure S9 – please show all repeats and the raw data, not just the fitted curves

Supplementary Figure S10a – please enlarge and increase font

Supplementary Figure S10b – please enlarge and provide labels

Supplementary Figure S11a, S12, 1d, 2e – please use a different color for each trace and label

Figure 2a, 2b, 2d, 3a-3d – please remove surface and provide sufficient labeling

Figure 3e – please replace the light blue with a bright red trace, please add molecular weights as calculated from polypeptide chain and as obtained from the experiment

Figure 4c – please show data and not just fitted curve for each repeat

Figure 4d and 4e – please provide the actual values also

Point-by-point response to the comments from Reviewer 1

We thank the reviewer for the careful reading of the paper and the constructive criticisms. We provide answers to all the comments as follows.

Comment from the reviewer

The one experiment I feel is missing is the SEC-MALS of the protein-hit 1 complex.

>Our response to the comment

We acknowledge the reviewer's opinion. We have tried the experiment, but unfortunately, we did not obtain the difference between the size of X dimer (MEC12) in the presence or absence of Hit 1 (**Fig. R1a**). The possible explanation of this is that, the sample is diluted upon the loading to the column by at least a few-fold, and the dilution of the sample caused the ligand to dissociate from the proteins. In FFF-MALS experiment, where the dilution rate is lower than in SEC-MALS, the elution time of X dimer with or without Hit 1 was almost the same (**Fig. R1b**). In my case, since the affinity of Hit 1 to the X dimer or monomeric P-cadherin is weak (mM to sub mM), it would be difficult to keep the protein and Hit 1 in contact in the column.

Fig. R1 SEC and FFF measurement. The elution time of X dimer with or without Hit 1 was the same in (a) SEC, and (b) FFF.

Comment from the reviewer

The Figures and crystallography can be improved.

Supplementary Table 1 (please move into the main text)

- The Rmerge and Rmeas seem odd for the last shell
- The signal to noise too low in the last shell
- The number of solvent molecules seem too low at the given resolutions
- Please provide MolProbity scores

>Our response to the comment

We have modified the **Supplementary Table 1** according to the comments by providing MolProbity scores (REC12-Hit 1; 2.58, MEC12-Hit 1; 2.42) and have moved the table to main text as **Table 1**. We adopted CC1/2 value as a criterion for cutting off the high-resolution limit, which is a more appropriate indicator than R values to estimate the significance of the observations with large distribution such as weak diffraction signals at high resolution (Philip R. Evans and Garib N. Murshudov, *Acta Cryst. Sect. D*, 69, 1204-1214, 2013). Because of this criterion, the Rmerge and Rmeas value got high and the signal to noise got low in the highest resolution shell.

In both structures, water molecules were modeled automatically by using “ordered_solvent=true” of phenix.refine. As for the low number of solvent molecules, we are not sure about the exact reason, but we are sure that for both crystals, the soaking process caused some damage to the crystal, which might have resulted in the small number of solvent molecules, because the control crystal structure without soaking had considerable number of solvent molecules.

[p. 13 “Revised_manuscript.pdf”]

New Table 1
(Revised)

Data collection	REC12-Hit 1	MEC12-Hit 1
PDB ID	7CMF	7CME
space group	C 1 2 1	P 2 ₁ 2 ₁ 2 ₁
unit cell dimensions		
a , b , c (Å)	74.3 40.8 72.6	79.8 99.1 107.9
α , β , γ (°)	90 97.4 90	90 90 90
wavelength (Å)	1.0000	1.0000
resolution (Å)*	36.85 - 2.30 (2.39 - 2.30)	45.04 - 2.45 (2.60 - 2.45)
R _{merge}	0.15 (1.09)	0.07 (1.56)
R _{meas}	0.19 (1.28)	0.08 (1.67)

$CC_{1/2}$	0.99 (0.53)	0.99 (0.54)
$\langle I/\sigma(I) \rangle$	7.80 (1.28)	19.21 (1.30)
completeness (%)	94.4 (97.3)	99.9 (99.8)
redundancy	3.1 (3.1)	7.4 (7.5)
Refinement statistics		
resolution (Å)	36.85-2.30	45.04–2.45
R_{work}	0.245	0.209
R_{free}	0.287	0.249
No. of non-hydrogen atoms	1664	3388
macromolecules	1626	3303
ligands	18	56
solvent	20	29
unique reflections	9268 (940)	31974 (3199)
Average B-factor (Å ²)	54.57	75.81
MolProbity score	2.58	2.42
R. M. S. deviations from ideal		
bonds (Å)	0.005	0.01
angles (°)	0.77	1.15
Ramachandran plot (%)		
Favored region	94.76	96.46
Allowed region	4.76	3.07
Outlier region	0.48	0.47

Comment from the reviewer

Supplementary Figure S2 and S7a-c – please provide all surrounding protein atoms
 Supplementary Figure S4 – please enlarge the Figure to fill the page and provide sufficient labels

Supplementary Figure S6a through S6c – please enlarge to fill the entire page so that one can read it and replace the yellow with black

Supplementary Figure S6g – please show the monomer without surface in black, enlarge Figure and provide enough labels

Supplementary Figure S7d-f, 2c – please remove colors and increase font

Supplementary Figure S8, S11b, 4a, 4b – please increase brightness (S13 and 5d perhaps also)

Supplementary Figure S9 – please show all repeats and the raw data, not just the fitted

curves

Supplementary Figure S10a – please enlarge and increase font

Supplementary Figure S10b – please enlarge and provide labels

Supplementary Figure S11a, S12, 1d, 2e – please use a different color for each trace and label

Figure 2a, 2b, 2d, 3a-3d – please remove surface and provide sufficient labeling

Figure 3e – please replace the light blue with a bright red trace, please add molecular weights as calculated from polypeptide chain and as obtained from the experiment

Figure 4c – please show data and not just fitted curve for each repeat

Figure 4d and 4e – please provide the actual values also

>Our response to the comment

We are grateful for the comments to make the quality of the figures better. For **Supplementary Figure S6a** through **S6c**, the figure is made from the screenshot, so difficult to modify the color; thus we changed the saturation of the colors so that the figure is more easily readable. For **Supplementary Figure S7d-f**, **Figure 2c**, these are screenshots from the FLEV software, so the colors cannot be removed. Also, for **Supplementary Figure S9** (new **Supplementary Figure S11**) and **Figure 4c** (new **Figure 4d**), the O.D.₆₅₀ trace is raw data. All the repeats of the experiments are shown in new **Supplementary Fig. S11**. Accordingly, the representative traces among the three repeats are newly shown as new **Fig. 4d**. The other points are modified accordingly. Please check them out in the revised manuscript.

Reviewer 2

Original comments from Reviewer 2

Summary

This paper reports a chemical fragment that disrupts dimerization of P-cadherin (cadherin 3). The authors use a screen for cadherin-binding drug-like fragments and show using structural and biophysical techniques that the top hit compound binds to P-cadherin and lowers the on-rate of dimer formation. They then use basic cell biology experiments to suggest that the novel hit fragment inhibits cadherin-mediated cell aggregation/adhesion. The paper describes a valuable small-molecule screen, which yielded a potentially useful tool fragment candidate, with important potential applications that should be of interest to the field of adhesion biology. However, additional cell biological data are required to strengthen the conclusions and relevance to cell adhesion.

Major points

1. The authors should better introduce the hypothesis for targeting the hydrophobic pocket by expanding their argument in the Introduction with a more detailed rationale (p. 4, lines 56–59). It would be helpful to define more precisely for the reader why S–S dimer inhibition may not be the most appropriate approach, as alluded to in the manuscript.

2. The explanation of the structural change in Hit 1-bound X dimer (p. 13, lines 168–175) is not clear to me and would benefit from further corroboration. What is the evidence that two Hit 1 molecules bind in the cavity (or do the authors mean one Hit 1 molecule in each of the respective chains of the dimer)?

What is the evidence that the Hit 1-bound dimer is in a metastable state? More data or more detailed explanations are required to support these conclusions.

3. The authors should provide evidence that only P-cadherin remains on the cell surface following trypsinisation, as claimed in the manuscript, as this is central to the principle of the P-cadherin-dependent cell aggregation assay.

4. Can the authors quantify the cell aggregation data from the multiple experimental replicates (Figs 4 and 5) to provide firmer cell biological conclusions?

5. Findings from the cell aggregation assay should be verified using an alternative, quantitative assay (e.g. quantify cell–cell aggregation by calculating particle size

distribution after incubation of cells in suspension).

6. It is not clear how cell area was determined in Fig. 4e – is this mean area of an average cell or is it the proportion of the plate covered by cells (i.e. confluency)? More methodological details are required to interpret this experiment.

Furthermore, the biological relevance of this assay is not clear: the association between cell area and cell–cell adhesion, in particular the relevance to P-cadherin-dependent cell adhesion, needs to be explained.

7. Can the authors explain why an order of magnitude higher concentration of Hit 1 is required to inhibit cell aggregation as compared to cell area?

8. The dosing range of phenyl-Hit 1 does not appear appropriate for the cell aggregation assay in Fig. 5d. A low dose at which aggregation is not inhibited should be used – the small aggregates at 62.5 μ M indicated by an arrow are insufficient to convincingly show that the cells were healthy and able to aggregate normally in this experiment. In addition, this experiment should be performed alongside a positive and a negative control, and also alongside the original Hit 1 fragment to enable accurate comparison.

Minor points

1. Please use exact n number for number of independent experiments tested in Figs 1d, 2e, 3e, 4a, 4b, 4c and 5d.
2. The different construct names (e.g. REC12, MEC12) are not intuitive so should be summarised in a figure panel or table.
3. The fourth and fifth sentences of the Introduction (p. 3, lines 34–41) were difficult to follow and their clarity should be improved.
4. Make it clearer in Fig. 1 that the arrowed numbers in parentheses (e.g. 2 \rightarrow 1) represent decreases in the number of fragment candidates.
5. The angstrom symbol was not reproduced in the manuscript.
6. The label font in Fig. 2c is too small to read.

Point-by-point response to the comments from Reviewer 2

We thank the reviewer for the constructive, and important criticisms especially regarding the meaning of Hit 1-bound X dimer structure and several cell-based experiments. To address the concerns from the reviewer, we have revised the manuscript as follows.

Major comment from the reviewer

1. The authors should better introduce the hypothesis for targeting the hydrophobic pocket by expanding their argument in the Introduction with a more detailed rationale (p. 4, lines 56–59). It would be helpful to define more precisely for the reader why S–S dimer inhibition may not be the most appropriate approach, as alluded to in the manuscript.

>Our response to the comment

This is actually a very important point when we think of how to regulate the dimerization of cadherin molecules. In the previously reported crystal structures (Ref 21: Kudo *et al.*, *Structure*, 2016), the hydrophobic pocket is occupied in both monomeric form (PDB ID; 4zmz), and X dimer (PDB ID; 4zmq) with its own or the other monomer's tryptophan residues respectively. This fact gives us an estimation that the hydrophobic pocket used in S-S dimerization is always occupied during the process from monomer to S-S dimer, thus causing difficulty in the ligand binding. According to this estimation, we revised the sentences in introduction as follows.

[p. 5, line 8 in “Revised_manuscript.pdf”]

(Original)

Based on this binding mode, several peptide mimetic ligands have been reported^{23–28}; however, no ligands that bind to the hydrophobic pocket have been identified, and the molecular basis of inhibition of homodimerization by these peptide mimetics has not been elucidated.

(Revised)

Based on this binding mode, several peptide mimetic ligands have been reported^{23–28}; however, no ligands that bind to the hydrophobic pocket have been identified probably because the hydrophobic pocket is always occupied with either tryptophan of its own or of the other monomer at any stage in the process of S-S dimerization²¹, resulting in a poor understanding of the molecular basis of inhibition of homodimerization by these peptide mimetics

Major comment from the reviewer

2. The explanation of the structural change in Hit 1-bound X dimer (p. 13, lines 168–175) is not clear to me and would benefit from further corroboration. What is the evidence that two Hit 1 molecules bind in the cavity (or do the authors mean one Hit 1 molecule in each of the respective chains of the dimer)?

>Our response to the comment

Here, our meaning was “one Hit 1 molecule in each of the respective chains of the dimer”. The binding cavity was the same as that observed in REC12-Hit 1 complex structure (monomeric P-cadherin and Hit 1 complex). We revised the sentences so that readers can more easily understand what is happening in the Hit 1-bound state X dimer.

[p. 15, line 1 in “Revised_manuscript.pdf”]

(Original)

Three independent sets of electron density that can be modeled as Hit 1 were found, two of them in the cavity around Y140, the other at the intersection of EC2 domains (Fig. 3b).

(Revised)

Three independent sets of electron density that can be modeled as Hit 1 were found, two of them in the cavity around Y140, the same cavity in Fig. 2a, and from each monomeric chain, the other at the intersection of EC2 domains (Fig. 3b)

Major comment from the reviewer

What is the evidence that the Hit 1-bound dimer is in a metastable state? More data or more detailed explanations are required to support these conclusions.

>Our response to the comment

This also is a very important point. We speculated in the original manuscript that the Hit 1-bound state of X dimer was in a metastable state in the process of X dimerization solely from the fact that one of the interactions from hot spot residues was disrupted in the crystal structure we obtained. In this revision, in order to verify this speculation, we performed molecular dynamics (MD) simulations using the crystal structure of apo-state X dimer (PDB ID; 4zmq). MD simulations can trace solution state dynamics of a protein, and we hypothesized that a metastable state in

the process of X dimerization should be included in conformations of apo-state X dimer. To represent conformations of apo-state X dimer, we focused on the formation of a hydrogen bond in the pairs of K14-A138' and K14'-A138, since this is the key interaction in the process of X dimerization, as evidenced by our biochemical experiments. Based on the MD trajectories, we calculated the distances between a hydrogen bond donor and an acceptor in the pairs of K14-A138' and K14'-A138, and summarized them as frequency histograms (**Fig. 3f**). As a result, we observed two peaks in the histograms; both of them corresponded to the distances of the interactions observed in the crystal structure of the Hit 1-bound X dimer (the dotted line in **Fig. 3f**). That is, in our MD simulations, the apo-state X dimer was able to assume conformations observed in the crystal structure of Hit 1-bound X dimer, as stable states, even without Hit 1. Based on this result, we have added in the revised manuscript the following sentences. Also related figures have been added in **Figure 3** as **Fig. 3f**. Because the MD simulations were conducted with the help of Dr. Daisuke Kuroda, he was added to the author list.

[p. 16, line 9 in “Revised_manuscript.pdf”]

(Added)

To confirm that the hydrogen bond between the side chain of K14 and the main chain of A138' is important for X dimerization as previously reported²¹, we used a size exclusion chromatography-multiangle light scattering (SEC-MALS) analysis. Indeed, the K14A mutant was mainly in monomer form (**Fig.3e**).

Since the key interaction between K14 and A138' was disrupted in the Hit 1-bound X dimer, we speculate that the structure of X dimer may represent one of the metastable states in the process of X dimerization. To prove the metastable states or solution-state dynamics of X dimer in the apo form, molecular dynamics (MD) simulation was performed using the apo-state structure of X dimer (PDB ID; 4zmq). Three independent simulations for 100 ns each were conducted and the convergence of the trajectories was confirmed by root mean square deviation (RMSD) of C α atoms (**Supplementary Fig. S9a**). We then computed the distance of hydrogen bond donor (NZ) and acceptor (O) in the pairs of K14-A138' and K14'-A138, respectively (**Fig. 3f**), demonstrating that the frequency distribution has two peaks, the one at 2.6 Å and the other at 4.8 Å. This asymmetric nature was observed in all the trajectories (**Supplementary Fig. S9b**). In the crystal structure of the Hit 1-bound state X dimer, the distance of the pairs of K14-A138' and K14'-A138 were 4.6 Å, and 2.6 Å, respectively. This result implies that apo-state dynamics of X dimer could have two

possible states in terms of the hydrogen bond formation, and both states correspond to the ones observed in the crystal structure of the Hit 1-bound state of X dimer. In other words, the apo-state X dimer was able to assume conformations close to the ones Hit 1-bound X dimer exhibited in the crystal structure, even without Hit 1 compound. These observations support our hypothesis that the Hit 1-bound state of X dimer observed in the crystal structure we obtained is a metastable state in the process of X dimerization. Together, these data suggest that Hit 1 binding may interfere with formation of hydrogen bonds necessary for X dimerization by trapping X dimer into the metastable state. Unlike a typical orthosteric inhibitor, Hit 1 did not directly block hydrogen bonding between two monomer units; rather, Hit 1 may alter the monomer structure to prevent hydrogen bond formation.

[p. 39 in “Revised_manuscript.pdf”]

(Added)

MD simulation

Molecular dynamics simulations were performed using GROMACS 2016.3⁵² with the CHARMM36m force field⁵³. A crystal structure of apo-state X dimer (PDB ID; 4zmq) was used as the initial coordinates. The protein was solvated with TIP3P water⁵⁴ in a rectangular box such that the minimum distance to the edge of the box was 15 Å under periodic boundary conditions through the CHARMM-GUI⁵⁵. Na⁺ or Cl⁻ ions were added to imitate a salt solution of concentration 0.15 M. Each system was energy-minimized for 5000 steps and equilibrated with the NVT ensemble (303 K) for 1 ns. Further simulations were performed with the NPT ensemble at 303 K. The time step was set to 2 fs throughout the simulations. A cutoff distance of 12 Å was used for Coulomb and van der Waals interactions. Long-range electrostatic interactions were evaluated using the particle mesh Ewald method⁵⁶. Covalent bonds involving hydrogen atoms were constrained by the LINCS algorithm⁵⁷. A snapshot was saved every 10 ps. All trajectories were analyzed using GROMACS with the converged trajectories from 40 ns to 100 ns (**Supplementary Fig.S9a**).

[p. 18 in “Revised_manuscript.pdf”]
 (New Figure 3)

Fig. 3 | Effects on X dimerization by Hit 1. (a) Hydrogen-deuterium exchange ratio of two peptides 134-140 and 137-147 as a function of time. The region of these peptides is colored in magenta in the structure of MEC12, X dimer (PDB ID; 4zmq). Chain A of X dimer is colored in sky blue; chain B in pink. (b) Structure of the complex of the X dimer and Hit 1. Hit 1 (in sphere, orange) is bound in the cavity around Y140 of each monomeric chain and at the intersection of EC2 domains. The pink or sky blue or black arrows show the Hit 1 that binds to chain A or chain B or at the intersection of EC2 domains respectively. (c) Structural change caused by the binding of Hit 1. Hit 1-bound X dimer is aligned to the apo X dimer (PDB ID; 4zmq, black). The angle of EC1 to EC2 is flatter compared to that in the apo X dimer (PDB ID; 4zmq). The arrows indicate movement of domains. (d) Hydrogen bonds disrupted by the structural change that occurs upon Hit 1 binding. Left: Apo X dimer with K14-A138' hydrogen bond is shown in dotted line. Right: Hit 1-bound X dimer; the K14-A138' hydrogen bond is disrupted. Amino acid residues from chain A are shown in sky blue, those from chain B are in pink. (e) Size measurement using SEC-MALS. The WT X dimer trace is in blue, the K14A mutant trace is in red. N=1. (f) Frequency histogram of the distance between a hydrogen bond donor and an acceptor of the pairs of K14-A138' and K14'-A138 observed in the MD simulations of the apo-state X dimer (PDB ID; 4zmq). The black dotted lines represent the distances observed in the crystal structure of Hit 1-bound X dimer we obtained in this study.

[p. 13 in “Revised_SI.docx”]
(New Supplementary Fig. S9)

Supplementary Fig. S9 | Time sequence of RMSD of the whole structure and a distance between two atoms of apo-state X dimer. (a) RMSD of $C\alpha$ atoms. (b) The

distance between a hydrogen bond donor and an acceptor in the pairs of K14-A138' and K14'-A138.

Major comment from the reviewer

3. The authors should provide evidence that only P-cadherin remains on the cell surface following trypsinisation, as claimed in the manuscript, as this is central to the principle of the P-cadherin-dependent cell aggregation assay.

>Our response to the comment

In the cell aggregation assay using the mock cells of Flp-In CHO, it cannot form cell aggregates more than 40 μm in size, which was revealed by the MFI experiment I am describing on the following page. Similarly, in the cell aggregation assay using the P-cadherin-expressing CHO cells, the larger cell aggregates were observed only in the condition with calcium, but not in the condition with EDTA. This indirectly indicates that in our cell aggregation assay, there is little, if any, extracellular protein other than P-cadherin and cell aggregation formation was basically dependent on the interactions among P-cadherin molecules. In order to let the readers more easily understand this, we added the sentences below. The related figure is also modified.

[p. 20, line 6 in “Revised_manuscript.pdf”]

(Added)

In this assay, extracellular proteins other than P-cadherin are trypsinized so that cell adhesion and formation of cell aggregates depend only on the interaction of P-cadherin molecules. **Indeed, the mock CHO cells did not form large cell aggregates (Fig. 4c).**

[p. 23 in “Revised_manuscript.pdf”]
 (New Figure 4)

Fig. 4 | Hit 1 blocks aggregation of liposomes and cells with P-cadherin on their surfaces. (a) Images of CHO cell line expressing P-cadherin in the presence of 1 mM CaCl₂ and 1 mM EDTA and mock cell line. Scale bars indicate 500 μm. (b) Representative images of CHO cell line expressing P-cadherin in the presence of 1 mM CaCl₂ and increasing concentrations of Hit 1. This experiment was repeated three times, and similar results were obtained. Scale bars indicate 500 μm. (c) Quantification of cell aggregation assay using MFI. Hit 1 inhibited the formation of cell aggregation in a dose-dependent manner. N=3. (d) Liposomes with MEC12 on the surface were incubated with CaCl₂ and with (red) or without (black) Hit 1. Controls had no effect on optical density (blue, yellow, and grey traces). Representative plot of optical density at 650 nm is shown as a function of time. N=3. (e) The k_{obs} of liposome aggregation calculated assuming a one-phase association

model. N=3. Error bars show standard deviation. Individual data points are shown in black plots. (f) Normalized area of HCT116 and MCF7 cells in the presence or absence Hit 1. In order to take cell shape into account, the cell area in the absence of Hit 1 was taken as 100%. N=3. Error bars show standard deviation. Individual data points are shown in black plots.

Major comment from the reviewer

4. Can the authors quantify the cell aggregation data from the multiple experimental replicates (Figs 4 and 5) to provide firmer cell biological conclusions?
5. Findings from the cell aggregation assay should be verified using an alternative, quantitative assay (e.g. quantify cell–cell aggregation by calculating particle size distribution after incubation of cells in suspension).

>Our response to the comment

We agree that this comment is also very important to make the result more meaningful. To address these comments, we established an assay system to quantify the cell aggregation assay, or cell aggregation disruption assay. We used Micro-Flow Imaging (MFI) instrument to count and measure the size distribution of the cell aggregates (new **Figure 4c**). The MFI instruments allowed us to count the number of particles 10-15 μm , 15-25 μm , 25-40 μm , 40-50 μm , 50-70 μm , and 70-100 μm respectively. Unfortunately, the number of cells included in each size of cell aggregates is unclear in this assay. Before the MFI measurement, 4% PFA was added to the final concentration of 0.8% and the cells were incubated at r.t. for an hour. Because the addition of PFA in the presence of 500 μM Hit 1 (or 250 μM phenyl-Hit 1) caused precipitation, which seriously affects the count of cell aggregates, the concentration range of the compounds is different in the revised version from that in the original version. As an evaluation, we compared the count of cell aggregated larger than 40 μm because a considerable number of cell aggregates smaller than 40 μm was observed in mock CHO cells or the condition with EDTA too. Also, to normalize the number of the counts in each experiment, the ratio of the total number of particles larger than 40 μm to 100 μm over that of the particles larger than 25 μm to 100 μm was calculated. The resulting phenomenon observed by MFI experiments were almost the same as the original version; 1) Hit 1 showed the inhibition of cell aggregation formation in a dose-dependent manner, while it did not disrupt the pre-formed cell aggregates (new **Figure 4c**, new **Supplementary Figure S10**), 2) Phenyl-Hit 1 showed stronger inhibitory activity than Hit 1 (new **Figure 5e**). For

some wells where cell aggregation formation is NOT inhibited, the error bars of the MFI experiment became large. This may be because of the poor dispersibility of the samples and difficulty in even sampling. Note that, from the cell aggregation disruption assay using EDTA, some amount of cell aggregated remained formed after the incubation in our experimental condition, which suggests that once cell aggregates are formed, they are relatively stable. Based on these findings, the related sentences were modified as follows. Please see new **Fig. 4** also.

[p. 20, line 6 in “Revised_manuscript.pdf”]

(Added)

The aggregation reaction was initiated by addition of 1 mM CaCl₂. After the cell aggregation reaction, the size distribution of cell aggregates was measured using micro flow imaging (MFI) instrument. In MFI measurement, the number of cell aggregates more than 40 μm in size was counted as P-cadherin-dependent cell aggregates, since considerable number of cell aggregates of the size 25 μm to 40 μm were observed in the presence of EDTA. In order to normalize the number of the counts, the ratio of the number of particles larger than 40 μm to 100 μm over that of the particles larger than 25 μm to 100 μm was calculated. In the control experiment, EDTA inhibited formation of cell aggregates; thus adhesion is based on the calcium-dependent interaction of P-cadherin molecules (Figure 4a, 4c). When Hit 1 and 1 mM CaCl₂ were added simultaneously, aggregation was inhibited in a manner that depended on the Hit 1 concentration (Fig. 4b, 4c). When Hit 1 was added to pre-formed cell aggregates, the aggregates remained stable in the time scale of this assay; EDTA partially disrupted pre-formed aggregates (Supplementary Fig. S10). Considering these results, it may be that Hit 1 blocks association of two monomers in the process of the X dimerization. and that once cell aggregates are formed probably based on *cis* clustering, it is too stable to disrupt with Hit 1.

[p. 41, line 13 in “Revised_manuscript.pdf”]

(Added)

Micro-Flow Imaging (MFI)

Micro-Flow Imaging (Brightwell Technologies) was used in order to measure the size distribution of cell aggregates. After the cell aggregation assay, the plate was incubated at room temperature for 10 minutes and 125 μL of chilled 4% Paraformaldehyde Phosphate Buffer Solution (Nacalai Tesque) was added to each well. The plate was incubated at r.t. for more than 1 h. After that, cells were injected

to MFI. After the MFI measurement, the number of cell aggregates larger than 40 μm in size was counted as P-cadherin-dependent cell aggregates.

Major comment from the reviewer

6. It is not clear how cell area was determined in Fig. 4e – is this mean area of an average cell or is it the proportion of the plate covered by cells (i.e. confluency)? More methodological details are required to interpret this experiment.

>Our response to the comment

The meaning of **Figure 4e** (new **Figure 4f**) was the proportion of the plate covered by cells. We added more detailed explanation of how the normalized cell area was determined and calculated as follows.

[p. 22, line 7 in “Revised_manuscript.pdf”]

(Original)

Cells were plated in the presence of 100 μM Hit 1 and cell adhesion was quantified by cell area.

(Added)

Cells were plated in the presence of 100 μM Hit 1 and the extent of cell adhesion was quantified indirectly by total cell area. Since the area from each single cell was different in HCT116 and MCF7, the normalized cell area defined as the total cell area in the presence of each concentration of Hit 1 over the total cell area in the absence Hit 1 was calculated.

Major comment from the reviewer

Furthermore, the biological relevance of this assay is not clear: the association between cell area and cell–cell adhesion, in particular the relevance to P-cadherin-dependent cell adhesion, needs to be explained.

>Our response to the comment

We thank the reviewer for encouraging us to discuss the result from a more biological viewpoint. In a previous report (Ref 17, Zang C. C., *et al.*, *Clin. Cancer Res.*, **16**, 5177, 2010, Lichao S. *et al.*, *Am. J. Pathol.*, **179**, 380, 2011), the downregulation of P-cadherin is known to lead growth and proliferation of colon cancer cells. In those reports, they assume that the knockdown of P-cadherin led such phenotype because

the expression level of β -catenin, an intracellular protein-protein interaction partner of P-cadherin, also becomes down-regulated, resulting in the apoptosis of cells. Together with our results, the inhibition of cell adhesion mediated by P-cadherin might have an effect like P-cadherin knockdown. Although the effect of Hit 1 on this signaling pathway has not been investigated in this study, our result does not become contradictory with those results. Therefore, as just speculation, the following sentences were added.

[p. 22, line 12 in “Revised_manuscript.pdf”]

(Original)

This result indicates that Hit 1 blocks adhesion of cancer cells that express P-cadherin.

(Revised)

In the previous reports, the effect of P-cadherin on cancer metastasis is discussed based on the knockdown of P-cadherin and the resulting impact on signaling pathway^{17, 36}. This result, together with the phenomenon from the previous studies, indicates that Hit 1 blocks adhesion of cancer cells that express P-cadherin, which may result in the down-regulation of β -catenin pathway and activation of apoptosis of cells^{17, 36}.

Major comment from the reviewer

7. Can the authors explain why an order of magnitude higher concentration of Hit 1 is required to inhibit cell aggregation as compared to cell area?

>Our response to the comment

After we quantified the inhibition of cell aggregation using MFI, the concentration range does not seem so different. At 100 μ M, approximately 50% of cell aggregation formation was inhibited and 20-30% of normalized cell area was decreased, which will not be so big a difference as magnitude higher concentration. Actually, we wanted to set the same concentration series for this assay as that in the cell aggregation assay, but the solubility of Hit 1 in 0.1% DMSO was not so high.

Major comment from the reviewer

8. The dosing range of phenyl-Hit 1 does not appear appropriate for the cell aggregation assay in Fig. 5d. A low dose at which aggregation is not inhibited should be used – the small aggregates at 62.5 μ M indicated by an arrow are insufficient to convincingly show

that the cells were healthy and able to aggregate normally in this experiment. In addition, this experiment should be performed alongside a positive and a negative control, and also alongside the original Hit 1 fragment to enable accurate comparison.

>Our response to the comment

We agree with this opinion. A series of experiments was repeated with the same lot of cells and inhibitory activity was quantified using MFI instrument. Because of the reason stated on p.18 of this response sheet, the inhibitory activity of both Hit 1 and phenyl-Hit 1 was compared in the concentration range below 100 μ M. All the experiments were performed alongside the control well where only calcium is added. The result we obtained was in line with the original version. Based on this experiment, we have modified **Figure 5**.

[p. 27 in “Revised_manuscript.pdf”]
 (New Figure 5)

Fig. 5 | Phenyl-Hit 1 more effectively inhibits aggregation than does Hit 1. (a), (b) Chemical structures of the compounds investigated. (c) The k_{obs} values for indicated compounds in the liposome aggregation assay. N=3. Error bars show standard deviations. Individual data points are shown in black plots. (d) Representative images from cell aggregation assay in the presence of a range of

concentrations of phenyl-Hit 1. Scale bars indicate 500 μm . $N = 3$. In order to compare the inhibitory activity of Hit 1 and phenyl-Hit 1, the same lot of cells was used for both compounds. (e) Quantification of cell aggregation assay. $N = 3$. *Significant difference ($p < 0.05$ by one-sided Welch's t -test. $p = 0.032$)

Minor comments from the reviewer

1. Please use exact n number for number of independent experiments tested in Figs 1d, 2e, 3e, 4a, 4b, 4c and 5d.
2. The different construct names (e.g. REC12, MEC12) are not intuitive so should be summarised in a figure panel or table.
3. The fourth and fifth sentences of the Introduction (p. 3, lines 34–41) were difficult to follow and their clarity should be improved.
4. Make it clearer in Fig. 1 that the arrowed numbers in parentheses (e.g. 2 \rightarrow 1) represent decreases in the number of fragment candidates.
5. The angstrom symbol was not reproduced in the manuscript.
6. The label font in Fig. 2c is too small to read.

>Our response to the comments

We thank the reviewer for carefully reading and checking the manuscript. We have modified the related figures according to the comments. Especially for the comment No. 3, we have modified the sentences as follows.

[p. 4, line 6 in “Revised_manuscript.pdf”]

(Original)

However, the fundamental interaction of the assembly formation; protein-protein interactions are difficult to regulate with small molecules for several reasons: large surface areas are usually involved in interactions between proteins, whereas the accessible surface areas of chemical ligands are small, there are generally no substantial grooves at the protein-protein interface, and there are few natural inhibitors of protein-protein interactions to guide ligand design.

(Revised)

However, the fundamental interaction of the assembly formation; protein-protein interactions are difficult to regulate with small molecules for several reasons;1) large surface areas are usually involved in interactions between proteins, whereas the accessible surface areas of chemical ligands are small, 2) there are generally no

substantial grooves at the protein-protein interface, and 3) there are few natural inhibitors of protein-protein interactions to guide ligand design

Reviewer 3

Original comments from Reviewer 3

In this study, the authors describe the identification and characterization of a compound that binds a region between P-cadherin domains EC1 and EC2 and blocks P-cadherin homo dimerization. They use several different approaches to characterize Hit 1 binding and inhibition of homo dimerization and cell adhesion, including mutagenesis, SPR, crystallography, mass spec, and light scattering. They speculate, based largely on the position of the binding site of Hit 1 that this compound acts by impeding the kinetics of homo dimerization.

Although the results are interesting and provide one mechanism for inhibiting cadherin binding, the argument regarding the mechanism of action is speculation because they don't actually show that the compound alters the binding kinetics, rather than the homo dimerization affinity. Some of the experiments also need additional controls. Detailed comments are given below.

1. All SPR measurements were done under equilibrium binding conditions and do not report the effect of Hit1 on P-cadherin dimerization kinetics or binding affinity. An alternative interpretation is that Hit1 allosterically alters the dimerization affinity. The authors need to quantify the influence of Hit1 on the cadherin dimerization affinity to rule this out and to support their assertion that Hit1 affects kinetics. SPR measurements of the dimerization kinetics +/- Hit1 are also needed to support their arguments.

For example, compare P-cadherin dimerization kinetics by immobilizing REC12 with Hit1 or low calcium to disrupt homo dimers, and then compare the rate and limiting amount of homo dimerization in the presence of Hit1.

2. Based on structure analysis, they mutated the Hit1 binding site, but did not test whether the Y140R mutation affects P-cadherin dimerization. This would support their claim on lines 116-117 that disrupting H-bonding around the binding site affects P-cadherin binding.

Secondary structure measurements do not address this because even calcium depletion does not completely disrupt the secondary structure, even though it blocks dimerization.

3. In liposome aggregation measurements to demonstrate kinetic effects, the authors add

calcium and Hit1 simultaneously. They attribute reduced aggregation kinetics to altered X-dimer kinetics. They do not actually show this. As performed in this study, liposome aggregation convolves protein refolding/activation and cadherin binding. An alternative explanation is that Hit1 impedes calcium binding and cadherin activation. They should show altered kinetics at the protein level

4. On line 148-149, what is MEC12? Clearly state what this protein is in the text.
5. Measure Hit1 binding to MEC12 and compare with REC12. Are the affinities different?
6. Line 174, what do they mean by 'metastable state'?
7. Based on the evidence, argument in lines 180-183 is speculation only.
8. The inability of Hit1 to disrupt cell aggregates does not prove the equilibrium is unaffected. This could be due to diffusion limitation in dense cell aggregates. The authors should show this at the protein level with SPR.
9. In most of their figures, lettering on graphs is too small (especially SPR data) and is barely readable.
10. Technical details: How much P-cadherin dimer is bound initially? If they immobilize in REC12 in low calcium solution or the presence of Hit1, does this change the amount of cadherin initially loaded on the chip? How does this affect their Hit1 affinity estimations?

Point-by-point response to the comments from Reviewer 3

We thank the reviewer for the constructive, and important criticisms especially regarding the concept of this study, SPR-based assays and interpretation of several assays. To address the concerns from the reviewer, we have revised the manuscript as follows.

Major comment from the reviewer

1. All SPR measurements were done under equilibrium binding conditions and do not report the effect of Hit1 on P-cadherin dimerization kinetics or binding affinity. An alternative interpretation is that Hit1 allosterically alters the dimerization affinity. The authors need to quantify the influence of Hit1 on the cadherin dimerization affinity to rule this out and to support their assertion that Hit1 affects kinetics. SPR measurements of the dimerization kinetics +/- Hit1 are also needed to support their arguments.

For example, compare P-cadherin dimerization kinetics by immobilizing REC12 with Hit1 or low calcium to disrupt homo dimers, and then compare the rate and limiting amount of homo dimerization in the presence of Hit1.

>Our response to the comment

This comment partially includes some misunderstanding of the original manuscript (REC12 is a construct that keeps monomeric P-cadherin), but still includes very important points. Although the kinetic effect of Hit 1 on X dimerization is indirectly assessed using the liposome aggregation assay, the direct evidence that Hit 1 affects the on-rate of X dimer is absent; it remains unclear whether the effect is on on-rate or off-rate. Therefore, we completely agree with this opinion of the reviewer's and tried to get the kinetics parameters of X dimerization in the presence or absence of Hit 1 using SPR. We immobilized MEC12 (the construct that can form only X dimer, but not S-S dimer) on the Sensor Chip CM5 and the same protein sample MEC12 (34 μ M ~ 2.125 μ M) was injected onto the sensor chip surface with or without 200 μ M Hit 1. We obtained the binding response, but the shape of sensorgram was not usual, resulting in poor fitting and failure to obtain reliable kinetics parameters (**Fig. R2**), although the on-rate looked slower in the presence of Hit 1. From this, we decided that the word "on-rate modulation" is an overestimation, and that we do not use the word. Accordingly, the title of this study was changed, and all of the words in the original manuscript were changed.

Fig. R2 | Multicycle kinetics of X dimer (MEC12). (a) in the presence or (b) in the absence of Hit 1. Blue color or orange color traces are the fitting to the raw data (black traces).

[p. 1 in “Revised_manuscript.pdf”]

(Original title)

On-rate modulation of cadherin interactions by chemical fragments

(New title)

Regulation of cadherin dimerization by chemical fragments as a trigger to inhibit cell adhesion

[p. 29, line 13 in “Revised_manuscript.pdf”]

(Original)

The fragments identified herein had an impact on kinetics; probably because the energy barrier to remove the bound ligand should be cleared to complete the X dimerization, although the precise activation energy of X dimerization with or without Hit 1 remains unclear.

(Revised)

The fragments identified herein had an impact on kinetics; probably by trapping X dimer to an incomplete, metastable state, although whether Hit 1 modulates on-rate or off-rate of X dimerization remains unclear.

[p. 32 in “Revised_manuscript.pdf”]

(New Figure 6)

Fig. 6 | Inhibition of cell adhesion by a chemical fragment. In the presence of chemical fragments, the angle between the EC1 and EC2 domains is not optimal for X dimerization; thus, the process of X dimerization was inhibited. The inhibition of initial step in the whole process towards cell adhesion can be strong enough to inhibit cell adhesion.

Major comment from the reviewer

2. Based on structure analysis, they mutated the Hit1 binding site, but did not test whether the Y140R mutation affects P-cadherin dimerization. This would support their claim on lines 116-117 that disrupting H-bonding around the binding site affects P-cadherin binding.

>Our response to the comment

First of all, this comment may also include a misunderstanding of our original manuscript. The crystal structure for which this mutation work was performed is REC12-Hit 1 complex, and REC12 is a construct that keeps monomeric P-cadherin. Therefore, whether the REC12 Y140R is a dimer or not does not make difference in our discussion. But we have tested whether the MEC12 Y140R keeps X dimer or not. As a result, SEC-MALS experiments demonstrated that MEC12 Y140R keeps X dimer, and does not make difference in dimerization (**Fig. R3**). Note that the molecular weight of MEC12 was 33.8 kDa at the same concentration.

Fig. R3 | SEC-MALS of MEC12 Y140R. Mw of the peak was 36 kDa.

Major comment from the reviewer

3. In liposome aggregation measurements to demonstrate kinetic effects, the authors add calcium and Hit1 simultaneously. They attribute reduced aggregation kinetics to altered X-dimer kinetics. They do not actually show this. As performed in this study, liposome aggregation convolves protein refolding/activation and cadherin binding. An alternative explanation is that Hit1 impedes calcium binding and cadherin activation. They should show altered kinetics at the protein level

>Our response to the comment

We totally agree with this opinion and tested whether the calcium-binding is not inhibited or not in the presence of Hit 1. In order to investigate this, we performed DSC measurement. We found that in the absence of calcium, the thermal stability of MEC12 molecule becomes unstable ($T_m = 39.8\text{ }^\circ\text{C}$) compared to the condition with calcium ($T_m = 54.3\text{ }^\circ\text{C}$). Using this phenomenon, whether Hit 1 inhibits the calcium-binding or not was checked by measuring the T_m value of MEC12 in the presence of Hit 1 in a buffer that contains calcium. As a result, the addition of Hit 1 did not change the T_m value, indicating that the calcium-binding was not affected by Hit 1 (new **Supplementary Figure 12**). Accordingly, the related sentences were added as follows.

[p. 21, line 17 in “Revised_manuscript.pdf”]

(Added)

Note that it was not the activation of cadherin molecules that Hit 1 inhibited, because calcium binding of cadherin molecules was not inhibited in the presence of Hit 1, which was shown by DSC measurement in the presence or absence of calcium or Hit 1 (Supplementary Fig. S12).

[p. 17 in “Revised_SI.docx”]
(New Supplementary Figure 12)

Supplementary Fig. S12 | DSC measurements of X dimer. (a) DSC measurement of X dimer with or without Hit 1. (b) DSC measurement of X dimer with or without calcium.

[p. 42, line 15, “Revised_manuscript.pdf”]

Differential Scanning Calorimetry (DSC). DSC measurement was performed using an PEAQ-DSC instrument (Malvern). Samples were prepared at 1.0 mg/mL in

10 mM HEPES (pH 7.5), 150 mM NaCl with or without 3 mM CaCl₂, 5% DMSO, 1 mM Hit 1. Samples were heated from 20 °C to 110 °C at a rate of 60 °C/h.

Major comment from the reviewer

4. On line 148-149, what is MEC12? Clearly state what this protein is in the text.

>Our response to the comment

MEC12 is a construct that forms only X dimer, but not S-S dimer, which was explained in the original manuscript (p. 14, line 8 in “Revised_manuscript.doc”). But we agree that these words can cause misunderstanding, we added some explanation to the original manuscript so that the readers can easily understand what MEC12 or REC12 (or EC12). Please see the revised manuscript.

Major comment from the reviewer

5. Measure Hit1 binding to MEC12 and compare with REC12. Are the affinities different?

>Our response to the comment

We measured the affinity of Hit 1 to MEC12 and REC12 using the fluorescence proximity sensing mode in switchSENSE technology (Andreas Langer, *et al.*, *Nat. Commun.*, **4**: 2099, 2013), because the kinetics measurement using SPR was not feasible for fast on-rate and off-rate upon the protein-ligand interaction. We repeated the measurement of switchSENSE technology twice, and obtained a similar tendency that the affinity of Hit 1 to MEC12 (X dimer) was stronger than to REC12 (monomeric P-cadherin) in a similar concentration range observed in SPR (1.16 mM or 1.43 mM to X dimer, 5.97 mM or 2.51 mM to monomer). Although the K_D value to REC12 is a little larger than that measured by SPR (**Fig. 2d**, 0.92 mM), we do not think it a significant difference. Therefore, the related sentence was added in the revised manuscript. And new supplementary Fig. S8 was added.

[p. 15, line 5 in “Revised_manuscript.pdf”]

(Added)

Since Hit 1 that binds to the cavity in X dimer forms more non-covalent interactions with the residues around the pocket, the affinity of Hit 1 to X dimer seemed stronger than to monomer, which was confirmed by switchSENSE technology (**Supplementary Figure S8**).

[p. 38, line 13 in “Revised_manuscript.pdf”]

(Added)

Kinetics measurement using switchSENSE

Kinetics parameters upon the binding of Hit 1 onto MEC12 (X dimer) or REC12 (monomer) were measured on a heliX+ instrument (Dynamic Biosensors GmbH, Martinsried, DE) using the switchSENSE technology⁵¹ in static fluorescence proximity sensing mode. Biotinylated MEC12 or REC12 construct was immobilized on the ADP-48-2-0 Biochip with the Biotin Capture Kit (HK-SA-1), using the prehybridized Adapter-1-Ra strand with streptavidin and prehybridized Adapter-2-Ra ligand-free strand as a reference. Hit 1 was prepared in 10 mM HEPES, 150 mM NaCl, 3 mM CaCl₂, 0.05% Tween20, 5% DMSO, pH 7.5 buffer in dilutions from 2 mM to 31.2 μ M, and injected onto the sensor chip at 200 μ L/min at 23 °C. The consecutive dissociation was done at 200 μ L/min at 23 °C for 60 sec. Kinetics parameters were calculated using the proprietary heliOS software with a buffer and real-time referencing.

[p. 12 in “Revised_SI.docx”]

(New Supplementary Fig. S8)

Supplementary Fig. S8 | Kinetics measurement of Hit 1. (a) Hit 1 onto X dimer, MEC12 and (b) to monomer, REC12 using switchSENSE technology.

Major comment from the reviewer

6. Line 174, what do they mean by ‘metastable state’?

>Our response to the comment

We thank the reviewer for the critical comment. Actually, a similar comment was given by Reviewer 1, which was “What is the evidence that the Hit 1-bound dimer is in a metastable state? More data or more detailed explanations are required to support these conclusions.” The following is the answer that we prepared for Reviewer 1.

>Our response to the comment

This also is a very important point. We speculated in the original manuscript that the Hit 1-bound state of X dimer was in a metastable state in the process of X dimerization solely from the fact that one of the interactions from hot spot residues was disrupted in the crystal structure we obtained. In this revision, in order to verify this speculation, we performed molecular dynamics (MD) simulations using the crystal structure of apo-state X dimer (PDB ID; 4zmq). MD simulations can trace solution state dynamics of a protein, and we hypothesized that a metastable state in the process of X dimerization should be included in conformational ensembles of apo-state X dimer. To represent conformational ensembles of apo-state X dimer, we focused on the formation of a hydrogen bond in the pairs of K14-A138’ and K14’-A138, since this is the key interaction in the process of X dimerization, as evidenced by our biochemical experiments. Based on the MD trajectories, we calculated the distances between a hydrogen bond donor and an acceptor in the pairs of K14-A138’ and K14’-A138, and summarized them as frequency histograms (**Fig. 3f**). As a result, we observed two peaks in the histograms; both of them corresponded to the distances of the interactions observed in the crystal structure of the Hit 1-bound X dimer (the dotted line in **Fig. 3f**). That is, in our MD simulations, the apo-state X dimer was able to assume conformations observed in the crystal structure of Hit 1-bound X dimer, as stable states, even without Hit 1. Based on this result, we have added in the revised manuscript the following sentences. Also related figures have been added in **Figure 3** as **Fig. 3f**. Because the MD simulations were conducted with the help of Dr. Daisuke Kuroda, he was added to the author list.

[p. 16, line 9 in “Revised_manuscript.pdf”]

(Added)

To confirm that the hydrogen bond between the side chain of K14 and the main chain of A138’ is important for X dimerization as previously reported²¹, we used a size

exclusion chromatography-multiangle light scattering (SEC-MALS) analysis. Indeed, the K14A mutant was mainly in monomer form (**Fig.3e**).

Since the key interaction between K14 and A138' was disrupted in the Hit 1-bound X dimer, we speculate that the structure of X dimer may represent one of the metastable states in the process of X dimerization. To prove the metastable states or solution-state dynamics of X dimer in the apo form, molecular dynamics (MD) simulation was performed using the apo-state structure of X dimer (PDB ID; 4zmq). Three independent simulations for 100 ns each were conducted and the convergence of the trajectories was confirmed by root mean square deviation (RMSD) of C α atoms (**Supplementary Fig. S9a**). We then computed the distance of hydrogen bond donor (NZ) and acceptor (O) in the pairs of K14-A138' and K14'-A138, respectively (**Fig. 3f**), demonstrating that the frequency distribution has two peaks, the one at 2.6 Å and the other at 4.8 Å. This asymmetric nature was observed in all the trajectories (**Supplementary Fig. S9b**). In the crystal structure of the Hit 1-bound state X dimer, the distance of the pairs of K14-A138' and K14'-A138 were 4.6 Å, and 2.6 Å, respectively. This result implies that apo-state dynamics of X dimer could have two possible states in terms of the hydrogen bond formation, and both states correspond to the ones observed in the crystal structure of the Hit 1-bound state of X dimer. In other words, the apo-state X dimer was able to assume conformations close to the ones Hit 1-bound X dimer exhibited in the crystal structure, even without Hit 1 compound. These observations support our hypothesis that the Hit 1-bound state of X dimer observed in the crystal structure we obtained is a metastable state in the process of X dimerization. Together, these data suggest that Hit 1 binding may interfere with formation of hydrogen bonds necessary for X dimerization by trapping X dimer into the metastable state. Unlike a typical orthosteric inhibitor, Hit 1 did not directly block hydrogen bonding between two monomer units; rather, Hit 1 may alter the monomer structure to prevent hydrogen bond formation.

[p. 39 in “Revised_manuscript.pdf”]

(Added)

MD simulation

Molecular dynamics simulations were performed using GROMACS 2016.3⁵² with the CHARMM36m force field⁵³. A crystal structure of apo-state X dimer (PDB ID; 4zmq) was used as the initial coordinates. The protein was solvated with TIP3P water⁵⁴ in a rectangular box such that the minimum distance to the edge of the box was 15 Å under periodic boundary conditions through the CHARMM-GUI⁵⁵. Na⁺ or

Cl⁻ ions were added to imitate a salt solution of concentration 0.15 M. Each system was energy-minimized for 5000 steps and equilibrated with the NVT ensemble (303 K) for 1 ns. Further simulations were performed with the NPT ensemble at 303 K. The time step was set to 2 fs throughout the simulations. A cutoff distance of 12 Å was used for Coulomb and van der Waals interactions. Long-range electrostatic interactions were evaluated using the particle mesh Ewald method⁵⁶. Covalent bonds involving hydrogen atoms were constrained by the LINCS algorithm⁵⁷. A snapshot was saved every 10 ps. All trajectories were analyzed using GROMACS with the converged trajectories from 40 ns to 100 ns (**Supplementary Fig.S9a**).

[p. 18 in “Revised_manuscript.pdf”]
 (New Figure 3)

Fig. 3 | Effects on X dimerization by Hit 1. (a) Hydrogen-deuterium exchange ratio of two peptides 134-140 and 137-147 as a function of time. The region of these peptides is colored in magenta in the structure of MEC12, X dimer (PDB ID; 4zmq). Chain A of X dimer is colored in sky blue; chain B in pink. (b) Structure of the complex of the X dimer and Hit 1. Hit 1 (in sphere, orange) is bound in the cavity around Y140 of each monomeric chain and at the intersection of EC2 domains. The pink or sky blue or black arrows show the Hit 1 that binds to chain A or chain B or at the intersection of EC2 domains respectively. (c) Structural change caused by the binding of Hit 1. Hit 1-bound X dimer is aligned to the apo X dimer (PDB ID; 4zmq, black). The angle of EC1 to EC2 is flatter compared to that in the apo X dimer (PDB ID; 4zmq). The arrows indicate movement of domains. (d) Hydrogen bonds disrupted by the structural change that occurs upon Hit 1 binding. Left: Apo X dimer with K14-A138' hydrogen bond is shown in dotted line. Right: Hit 1-bound X dimer; the K14-A138' hydrogen bond is disrupted. Amino acid residues from chain A are shown in sky blue, those from chain B are in pink. (e) Size measurement using SEC-MALS. The WT X dimer trace is in blue, the K14A mutant trace is in red. N=1. (f) Frequency histogram of the distance between a hydrogen bond donor and an acceptor of the pairs of K14-A138' and K14'-A138 observed in the MD simulations of the apo-state X dimer (PDB ID; 4zmq). The black dotted lines represent the distances observed in the crystal structure of Hit 1-bound X dimer we obtained in this study.

[p. 13 in “Revised_SI.docx”]
(New Supplementary Fig. S9)

Supplementary Fig. S9 | Time sequence of RMSD of the whole structure and a distance between two atoms of apo-state X dimer. (a) RMSD of $C\alpha$ atoms. (b) The

distance between a hydrogen bond donor and an acceptor in the pairs of K14-A138' and K14'-A138.

Major comment from the reviewer

7. Based on the evidence, argument in lines 180-183 is speculation only.

>Our response to the comment

We agree with your opinion. In order to tone down these sentences, we revised the related sentences as follows.

[p. 17, line 7 in "Revised_manuscript.pdf"]

(Original)

Together, these data suggest that Hit 1 binding interferes with formation of hydrogen bonds necessary for X dimerization. Unlike a typical orthosteric inhibitor, Hit 1 did not directly block hydrogen bonding between two monomer units; rather, Hit 1 alters the monomer structure to prevent hydrogen bond formation.

(Revised)

Together, these data suggest that Hit 1 binding **may** interfere with formation of hydrogen bonds necessary for X dimerization **by trapping X dimer into a metastable state**. Unlike a typical orthosteric inhibitor, Hit 1 did not directly block hydrogen bonding between two monomer units; rather, Hit 1 **may** alter the monomer structure to prevent hydrogen bond formation.

Major comment from the reviewer

8. The inability of Hit1 to disrupt cell aggregates does not prove the equilibrium is unaffected. This could be due to diffusion limitation in dense cell aggregates. The authors should show this at the protein level with SPR.

>Our response to the comment

We do not think that the diffusion effect is significant especially for the small molecules, because one of the previous studies reported anti P-cadherin single chain Fv antibody (scFv) was able to disrupt the pre-formed cell aggregates (Kudo *et al.*, *Sci. Rep.*, **7**, 39518, 2017). Indeed, this comment of the reviewers gave us an opportunity to have a second thought on what we wanted to emphasize in the relevant

section. It is not whether the equilibrium between monomer and X dimer was affected or not *in vitro*, but rather that the formation of cell aggregates (*cis*-clustering) could be too stable a process, if not irreversible, so that a small molecule should target the initial step (X dimerization). The process of cell adhesion is known to consist of several steps; X dimerization, S-S dimerization, and *cis*-clustering (**Figure 1a**). Our results from the cell aggregation disruption assay suggest that once *cis*-clustering is formed, it is too stable to disrupt with a small molecule, but that X dimerization could be targeted with a small molecule. Still, this is also none other than speculation. Therefore, in order to tone down the phrase, the relevant sentences and figures were modified as follows.

[p. 21, line 1 in “Revised_manuscript.pdf”]

(Original)

These results suggest that Hit 1 does not shift the equilibrium to the monomer state at the cellular level. We hypothesized that Hit 1 blocks association of two monomers by affecting the on-rate of X dimerization.

(Revised)

Considering these results, it may be that Hit 1 blocks association of two monomers in the process of the X dimerization. and that once cell aggregates are formed probably based on *cis*-clustering, it is too stable to disrupt with Hit 1.

[p. 30, line 5 in “Revised_manuscript.pdf”]

(Original)

The most effective small-molecule inhibitor of the formation of a macromolecule complex should block the initial association of the component proteins.

(Revised)

In addition, the scFv, which had a nM range affinity to monomeric P-cadherin, was able to disrupt the pre-formed cell aggregates, whereas our chemical fragments were not. The cell aggregation disruption assay may suggest that cell aggregates are too stable to be regulated with a small molecule with a weak affinity. Therefore, the most effective small-molecule inhibitor of the formation of a **stable** macromolecule complex should block the initial association of the component proteins.

Major comment from the reviewer

9. In most of their figures, lettering on graphs is too small (especially SPR data) and is barely readable.

>Our response to the comment

We thank the reviewer for carefully reading the manuscript. We modified the figures accordingly. Please check the revised manuscript.

Major comment from the reviewer

10. Technical details: How much P-cadherin dimer is bound initially? If they immobilize in REC12 in low calcium solution or the presence of Hit1, does this change the amount of cadherin initially loaded on the chip? How does this affect their Hit1 affinity estimations?

>Our response to the comment

This comment also includes a misunderstanding, since REC12 is a construct that keeps monomeric P-cadherin. Besides, in this study, REC12 was immobilized on Sensor Chip SA via biotin-streptavidin interaction, which is not affected whether or not calcium is binding.

Other changes in the manuscript

We would like to change Acknowledgement, Data availability, and Author contribution due to the experiments related to this revision as follows.

[p. 43, line 13 in “Revised_manuscript.pdf”]

Acknowledgements

We thank Thermo Fisher Scientific for the technical support in HDX-MS experiments. We thank Dynamic Biosensors for the technical support in heliX measurement. The super-computing resource was provided by Human Genome Center, the Institute of Medical Science, the University of Tokyo. This work was supported by JSPS Grant-in-Aid for Scientific Research on Innovative Areas “Bio-metal” (19H05766 to K.T.), JSPS Grants-in-Aid for Scientific Research (16H02420 to K.T.), Platform Project for Supporting Drug Discovery and Life Science Research (Basis for Supporting Innovative Drug Discovery and Life Science Research (BINDS)) from Japan Agency for Medical Research and Development (AMED) (JP19am0101094 to K.T.), and a Grant-in-Aid for JSPS fellows (A.S.). The synchrotron radiation experiments were performed at BL26B2 of SPring-8 and were supported by BINDS from AMED (JP19am0101070). The fragment compounds were provided by Drug Discovery Initiative supported by BINDS from AMED (JP19am0101086). This work was partly supported by the World-leading Innovative Graduate Study Program for Life Science and Technology, The University of Tokyo, as part of the WISE Program (Doctoral Program for World-leading Innovative & Smart Education), MEXT, Japan.

Data Availability statements

Coordinates and structure factors for the structures are deposited in the Protein Data Bank under accession code 7CME and 7CMF. The authors declare that the data supporting the findings of this study are available within the paper and its supplementary information files.

Author contributions

A.S., S.N., S.K., T.T., and K.T. designed experiments, analyzed and discussed the results, and approved the manuscript. A.S., K.Y., S.I, G.U., performed crystallization experiments, and processed and determined the crystal structure. A.S. and Y.S. synthesized Hit 1, Hit 1-carboxylic acid and phenyl-Hit 1 with input from S.S.. A.S. performed MD simulations with input from D.K.. A.S., D.K. and S.N. wrote the

manuscript with input from K.T..

Reviewers' comments:

Reviewer #1 (Remarks to the Author):

The SEC-MALS data that are provided in the rebuttal add more doubt to the effects of Hit 1 on dimerization

Further, it seems that these are SEC data, not SEC-MALS, but even on a regular SEC, the difference of a dimer and a monomer should be obvious irrelevant of the elongated shape of the molecule

I do not follow the explanation that the sample is too dilute since dilution should not affect the oligomeric state

If Hit 1 dissociated indeed dissociated (at an affinity of 0.9 mM that the authors call "unreliable") then one wonders about the significance of this interactions and it's identification in the crystal and how do we know this density is not from the crystallization solution

The 2mFo-DFc map (Figure 2) must be shaved (a difference omit map would be better) although it is not stated that it is, and the views (not in stereo) are all side views with probably a too high contour of 1 r.m.s.d. which makes it difficult to judge

Further, in PDB entry 4zmz, Y140 is near a symmetry related molecule and a discussion of the crystal contacts seems missing in this manuscript

Also, if I am understanding this correctly, the manuscript presents the crystal structures of MEC12 which is "a construct that forms the X dimer but not the S-S dimer" but we are not told what mutations this entails and REC12 which is a "monomer mutant" (that one needs to dig into the literature to see what mutations are made)

The biochemistry is also performed on these mutants

Why not use the physiologically wild type instead?

With regards to the crystal structures, no improvements were made and I do not see how soaking would reduce the number of solvent molecules

Reviewer #2 (Remarks to the Author):

The authors have done a thorough job of addressing my comments.

A few minor comments resulting from the authors' revisions that should be addressed:

1. For the new Fig. 4c and Fig. 5e, use en-dashes (not tildes) to indicate particle size ranges on the y-axis labels. Also, the "(-)" sign after on the y-axis labels (presumably to indicate no units) is ambiguous, so best removed.
2. For the new text referring to Fig. 4c (p. 20), it would be more appropriate to state that EDTA inhibited formation of "large cell aggregates (40 μm to 100 μm)" or similar.
3. For the revised text in the Introduction (p. 4), in the first sentence I mentioned in my original review (now lines 5–9), the first semi-colon (line 5) does not seem appropriate and makes the sentence difficult to follow. In the second sentence I mentioned (now lines 11–12), which has

remained unchanged, the meaning is still not clear, and this should be revised to improve clarity.

4. The font size used in Fig. 2c is still too small to read and will not reproduce well in production.

Response to the referees' comments and revisions that have been made

We thank the editor and reviewers for the constructive comments on our manuscript. We revised the manuscript in light of all the comments. Please check the following point-by-point response to the reviewers' comments as well as attached the revised files ("2nd_Revised_manuscript.pdf", and "2nd_Revised_SI.pdf"). We also attached copies of the revised files showing track changes for the benefit of the reviewers. We hope these answers would resolve all the reviewers' concerns.

(black: comments from the reviewers, blue: our comments, red: inserted/added/modified sentences/figures/tables)

Reviewer 1

Original comments from Reviewer 1

Reviewer #1 (Remarks to the Author):

The SEC-MALS data that are provided in the rebuttal add more doubt to the effects of Hit 1 on dimerization

Further, it seems that these are SEC data, not SEC-MALS, but even on a regular SEC, the difference of a dimer and a monomer should be obvious irrelevant of the elongated shape of the molecule

I do not follow the explanation that the sample is too dilute since dilution should not affect the oligomeric state

If Hit 1 dissociated indeed dissociated (at an affinity of 0.9 mM that the authors call “unreliable”) then one wonders about the significance of this interactions and it’s identification in the crystal and how do we know this density is not from the crystallization solution

The 2mFo-DFc map (Figure 2) must be shaved (a difference omit map would be better) although it is not stated that it is, and the views (not in stereo) are all side views with probably a too high contour of 1 r.m.s.d. which makes it difficult to judge

Further, in PDB entry 4zmz, Y140 is near a symmetry related molecule and a discussion of the crystal contacts seems missing in this manuscript

Also, if I am understanding this correctly, the manuscript presents the crystal structures of MEC12 which is “a construct that forms the X dimer but not the S-S dimer” but we are not told what mutations this entails and REC12 which is a “monomer mutant” (that one needs to dig into the literature to see what mutations are made)

The biochemistry is also performed on these mutants

Why not use the physiologically wild type instead?

With regards to the crystal structures, no improvements were made and I do not see how soaking would reduce the number of solvent molecules

Point-by-point response to the comments from Reviewer 1

We thank the reviewer for the constructive criticisms. We provide answers to all the comments as follows.

Comment from the reviewer

The SEC-MALS data that are provided in the rebuttal add more doubt to the effects of Hit 1 on dimerization

Further, it seems that these are SEC data, not SEC-MALS, but even on a regular SEC, the difference of a dimer and a monomer should be obvious irrelevant of the elongated shape of the molecule

>Our response to the comment

We deeply appreciate the important comment. We agree that the SEC (or SEC-MALS) experiment can distinguish X dimer and monomeric state; while the result that we provided shows that Hit 1 did not change the oligomeric state of X dimer under our experimental conditions, whether because of dilution of the sample or not.

We do not make much of whether Hit 1 actually shifts the equilibrium of X dimer or not. Rather, we put an emphasis on the fact that Hit 1 has an impact on kinetics, the process towards the final state of X dimer. In the extreme, a compound that makes the on-rate of X dimer slower, makes the off-rate of X dimer slower, and thus does not change the K_D of X dimer could be an effective inhibitor for cell adhesion. That is why we performed liposome aggregation assay starting from monomer to X dimer (please see p.22 line 1-5 in the original manuscript). The assay that should be adopted in the story of our manuscript is not the ones to check the final equilibrium state of X dimer like SEC (or SEC-MALS), but the ones that can trace the process of X dimerization like liposome aggregation assay or SPR (In the first revision, we attached SPR measurement on the point-by-point response sheet as **Fig. R2** as another result to suggest the decreased on-rate of X dimerization, which we decided not to use due to the poor fitting). The liposome aggregation assay (**Fig. 4d, 4e**) suggests that Hit 1 regulates the kinetics, probably on-rate of X dimerization. However, since the liposome aggregation assay does not give us precise information on off-rate from X dimer to monomeric state, we cannot discuss the impact of Hit 1 on K_D (or the final equilibrium state) of X dimerization from the experiment. This observation that kinetics of X dimerization could be an important factor to regulate the cell adhesion is what we want to emphasize in this manuscript consistently from

the beginning. Nonetheless, in the original manuscript, we mentioned that the equilibrium between monomer and X dimer was shifted to the monomeric state in the presence of Hit 1 as an interpretation of results from HDX-MS, which we came to think to be very confusing or even inaccurate against our conclusion. Accordingly, we carefully reviewed the manuscript and revised the relevant sentences as follows.

(Original manuscript)

This result suggests that the equilibrium between monomer and X dimer shifted toward the monomer form in the presence of Hit 1

(Revised p.15 line 11 of 2nd_Revised_manuscript.pdf)

This result suggests that Hit 1 has some impact on the interface of X dimerization

(Original manuscript)

Hit 1 modulated the kinetics of X dimerization, and despite the lack of a strong thermodynamic effect, inhibited cell adhesion.

(Revised p.31 line 1 of 2nd_Revised_manuscript.pdf)

Hit 1 modulated the process towards X dimerization, and despite the lack of a strong effect that can directly and thermodynamically change the final equilibrium state of X dimer, may have inhibited cell adhesion.

Comment from the reviewer

I do not follow the explanation that the sample is too dilute since dilution should not affect the oligomeric state

>Our response to the comment

In our opinion, once the sample is injected to the system of SEC-MALS, the sample becomes diluted by a few times before the sample gets to the column. Since we cannot add Hit 1 at sufficiently high concentration, we thought that Hit 1 had dissociated from the protein at the moment of injection to the system before the protein reached the column.

Comment from the reviewer

If Hit 1 dissociated indeed dissociated (at an affinity of 0.9 mM that the authors call “unreliable”) then one wonders about the significance of this interactions and it’s identification in the crystal and how do we know this density is not from the

crystallization solution

>Our response to the comment

We think it to be an overestimation to determine the binding of Hit 1 only from crystal structure. That is why we performed the mutation study in SPR. To emphasize that we discuss the binding of Hit 1 using both structural and mutational study, the relevant sentences were revised as follows.

(Original manuscript)

Electron density corresponding to Hit 1 was observed inside a shallow cavity located between the EC1 and EC2 domains.

(Revised p.10 line 5 of 2nd_Revised_manuscript.pdf)

Electron density **that can be modeled as** Hit 1 was observed inside a shallow cavity located between the EC1 and EC2 domains.

(Original manuscript)

In order to confirm the binding mode observed in the crystal structure, we performed a SPR-based direct binding assay using REC12 Y140R.

(Revised p.10 line 12 of 2nd_Revised_manuscript.pdf)

In order to confirm the **relevant electron density** observed in the crystal structure **reflects the binding of Hit 1 in solution condition**, we performed a SPR-based direct binding assay using REC12 Y140R.

Comment from the reviewer

The 2mFo-DFc map (Figure 2) must be shaved (a difference omit map would be better) although it is not stated that it is, and the views (not in stereo) are all side views with probably a too high contour of 1 r.m.s.d. which makes it difficult to judge

>Our response to the comment

We added another snapshot of 2mFo-DFc map of Hit 1 from another angle in **Supplementary Figure 2**, where we showed 2mFo-DFc map at 0.9 σ level.

(Original Supplementary Figure 2)

(New Supplementary Figure 2)

Supplementary Fig. S2 | 2mFo-DFc map of Hit 1. 2mFo-DFc map for Hit 1 from REC12-Hit 1 complex at 0.9σ level is shown.

Comment from the reviewer

Further, in PDB entry 4zmq, Y140 is near a symmetry related molecule and a discussion of the crystal contacts seems missing in this manuscript

>Our response to the comment

The structures in Figure 2 and 4zmq are highly isomorphic; they both have almost the same unit cell and the same space group. We have also confirmed that Y140 and the area around the ligand binding site are indeed in the vicinity of the position associated with the symmetry operation, but they do not interact specifically with the neighboring molecules associated with it. In other words, we concluded that we only need to consider molecules within the asymmetric unit to discuss the ligand interactions. Therefore, we ignored the interactions that you pointed out. In conclusion, the difference between both structures should not arise from the packing effect. Accordingly, we added the sentence.

(Added p.10 line8)

Note that both structures with or without Hit 1 are highly isomorphic, so that the difference should not arise from packing effect.

Comment from the reviewer

Also, if I am understanding this correctly, the manuscript presents the crystal structures of MEC12 which is “a construct that forms the X dimer but not the S-S dimer” but we are not told what mutations this entails and REC12 which is a “monomer mutant” (that one needs to dig into the literature to see what mutations are made)

>Our response to the comment

Thank you very much for pointing out the ambiguous parts in the manuscript. We modified the relevant sentences accordingly.

(Added p.6 line 5 of 2nd_Revised_manuscript.pdf)

Monomer mutant of P-cadherin called REC12²¹, where arginine is inserted at the N termini of EC12, was immobilized

(Added p.15 line8 of 2nd_Revised_manuscript.pdf)

MEC12²¹, a construct that forms the X dimer but not the S-S dimer due to the inserted methionine at the N termini of EC12 to inhibit the strand-swapping,

Comment from the reviewer

The biochemistry is also performed on these mutants

Why not use the physiologically wild type instead?

>Our response to the comment

We also thought it to be more persuading to use the wild type (EC12) construct, which is in equilibrium between monomer and S-S dimer, for biochemistry experiments like liposome aggregation assay or HDX-MS. We tried them.

For the liposome aggregation assay, monomeric form of EC12 must be prepared to start the aggregation reaction by the addition of Ca^{2+} . To achieve that, we tried two things. First, we added EDTA to the purified S-S dimer to get the monomeric state; but we found that, once S-S dimer forms in the presence of calcium during the purification, it cannot get back to monomeric state even in the presence of EDTA, probably because EDTA inhibits the formation of X dimer in the process from S-S dimer to monomer (**Fig. R1a**). Second, we also attempted to purify the wild type (EC12) construct in the absence of calcium from the beginning, but we failed to do that due to lots of protein aggregation (**Fig. R1b**). For these reasons, we regret to say that the liposome aggregation assay using the wild type construct is difficult.

Fig. R1 SEC profiles regarding the revision experiments. (a) EC12 with or without EDTA. Orange trace; EC12 without EDTA, blue trace; EC12 with EDTA. Both samples eluted the same position shown by a black square, indicating that S-S dimer keeps dimeric form in the presence of EDTA. (b) Purification of EC12 without Ca^{2+} . Without Ca^{2+} , almost all of the protein solution after His-column purification eluted as a large void peak. The black arrow shows the theoretical elution volume for EC12 in monomeric form. Note that a peak around 180 mL to 200 mL is not derived from cadherin.

For HDX-MS, we performed the similar assay as shown in **Fig. 3a** using EC12 (wild type, forming S-S dimer) and REC12 (monomer), and attempted to discern the difference between monomer and S-S dimer. From the crystal structure of S-S dimer (PDB ID; 4zml), the residues reflecting the interface of S-S dimer are known to be

1st to 6th, and 22nd to 29th amino acids (**Fig. R2a**); therefore, the hydrogen-deuterium exchange ratio of the peptides that include these residues (“1 to 6” and “6 to 34”) are checked. We had expected that in these peptides, the deuterium exchange ratio of monomer was higher enough than that of S-S dimer just like **Fig. 3a**, but the hydrogen-deuterium exchange ratio of monomer was lower than that of S-S dimer in the peptide of “1 to 6”, or was almost identical to that in the peptide of “6 to 34” (**Fig. R2b**). These results suggest that the difference between monomer and S-S dimer is difficult to see in HDX-MS; thus difficulty in checking the effect of compound, too. Possible reason for this is that, the region of dimer interface in S-S dimer do not seem to be sufficiently exposed to the solvent even in monomeric state, since in monomeric state, the N terminal strand of its own is stored in the pocket comprised by the residues of dimer interface as described on p.5 line 8 of the revised manuscript.

Fig. R2 HDX-MS experiments using the S-S dimer (EC12) and monomer (REC12). (a) Crystal structure of EC12 (PDB ID; 4zml). The residues (1 to 6, 22 to 29) comprising the dimer interface are colored cyan. (b) The deuterium exchange ratio of two peptides (1 to 36, 6 to 34) as a function of time. Unlike the interface of X dimer (MEC12, **Fig. 3a**), the interface of S-S dimer does not seem to be exposed enough to the solvent and looks difficult to be monitored by this method.

Comment from the reviewer

With regards to the crystal structures, no improvements were made and I do not see how soaking would reduce the number of solvent molecules

>Our response to the comment

We discussed this issue thoroughly. We adopted a soaking method to obtain this complex structure. At this time, the osmotic shock by the soaking operation reduced the quality of the crystal to some extent. We do not think that the soaking process itself reduced the number of water molecules. There should have been a number of water molecules in the crystal, but the reduced crystal quality made it difficult for the modeling of water molecules by phenix.refine with resultant data.

In that sense, reviewer 1 is right in pointing out that there are not enough water molecules for the resolution of the crystal structure obtained. However, as we mentioned before, we adopted CC1/2 as a cutoff for high resolution. In fact, because of the ambiguity in the definition of the highest resolution and the differences in the way crystallographers define the highest resolution (this is still being discussed by many crystallographers, but we don't think there is a unified view yet), we simply adopted the commonly used criteria.

While the water molecule-derived electron density became less observable by soaking, the ligand-derived electron density was still visible. Paradoxically, we believe that this is evidence that the ligand binds to the protein in the crystal with a certain binding force. We also performed mutation work as a corroboration, and confirmed that the ligand binds to the binding site formed by Y140 and other residues

Regarding this comment of reviewer 1's, if possible, we would appreciate it if reviewer 1 could show us the clearer intention of the comment so that we could appropriately improve the crystal structure and address this comment. We attached the previous comment from the reviewer 1 regarding crystal structure and our answer to the comments below for the benefit of the reviewer 1.

Comment from reviewer 1 in the previous revision

The Figures and crystallography can be improved.

Supplementary Table 1 (please move into the main text)

- The Rmerge and Rmeas seem odd for the last shell
- The signal to noise too low in the last shell
- The number of solvent molecules seem too low at the given resolutions
- Please provide MolProbity scores

>Our response to the comment

We adopted CC1/2 value as a criterion for cutting off the high-resolution limit, which is a more appropriate indicator than R values to estimate the significance of the

observations with large distribution such as weak diffraction signals at high resolution (Philip R. Evans and Garib N. Murshudov, *Acta Cryst. Sect. D*, 69, 1204-1214, 2013). Because of this criterion, the Rmerge and R meas value got high and the signal to noise got low in the highest resolution shell.

In both structures, water molecules were modeled automatically by using “ordered_solvent=true” of phenix.refine. As for the low number of solvent molecules, we are not sure about the exact reason, but we are sure that for both crystals, the soaking process caused some damage to the crystal, which might have resulted in the small number of solvent molecules, because the control crystal structure without soaking had considerable number of solvent molecules.

Reviewer 2

Original comments from Reviewer 2

Reviewer #2 (Remarks to the Author):

The authors have done a thorough job of addressing my comments.

A few minor comments resulting from the authors' revisions that should be addressed:

1. For the new Fig. 4c and Fig. 5e, use en-dashes (not tildes) to indicate particle size ranges on the y-axis labels. Also, the “(-)” sign after on the y-axis labels (presumably to indicate no units) is ambiguous, so best removed.
2. For the new text referring to Fig. 4c (p. 20), it would be more appropriate to state that EDTA inhibited formation of “large cell aggregates (40 μm to 100 μm)” or similar.
3. For the revised text in the Introduction (p. 4), in the first sentence I mentioned in my original review (now lines 5–9), the first semi-colon (line 5) does not seem appropriate and makes the sentence difficult to follow. In the second sentence I mentioned (now lines 11–12), which has remained unchanged, the meaning is still not clear, and this should be revised to improve clarity.
4. The font size used in Fig. 2c is still too small to read and will not reproduce well in production.

Point-by-point response to the comments from Reviewer 2

We thank the reviewer for careful reading. To address the concerns from the reviewer, we have revised the manuscript as follows.

Comment from the reviewer

1. For the new Fig. 4c and Fig. 5e, use en-dashes (not tildes) to indicate particle size ranges on the y-axis labels. Also, the “(-)” sign after on the y-axis labels (presumably to indicate no units) is ambiguous, so best removed.

>Our response to the comment

We modified the figure accordingly. Please check the revised manuscript.

Comment from the reviewer

2. For the new text referring to Fig. 4c (p. 20), it would be more appropriate to state that EDTA inhibited formation of “large cell aggregates (40 μm to 100 μm)” or similar.

>Our response to the comment

We modified the figure accordingly.

(Original manuscript)

EDTA inhibited formation of cell aggregates

(Revised p.21, line 14 of 2nd_recised_manuscript.pdf)

EDTA inhibited formation of large cell aggregates (40 μm to 100 μm)

Comment from the reviewer

3. For the revised text in the Introduction (p. 4), in the first sentence I mentioned in my original review (now lines 5–9), the first semi-colon (line 5) does not seem appropriate and makes the sentence difficult to follow. In the second sentence I mentioned (now lines 11–12), which has remained unchanged, the meaning is still not clear, and this should be revised to improve clarity.

>Our response to the comment

We appreciate the comment. We modified the sentences as follows.

(Original manuscript)

However, the fundamental interaction of the assembly formation; protein-protein interactions are difficult to regulate with small molecules for several reasons; 1) large surface areas are usually involved in interactions between proteins, whereas the accessible surface areas of chemical ligands are small, 2) there are generally no

substantial grooves at the protein-protein interface, and 3) there are few natural inhibitors of protein-protein interactions to guide ligand design³⁻⁵.

One of the protein assembly formation can be found in the process of forming cell adhesion by classical cadherin family proteins, calcium-dependent cell adhesive molecules.

(Revised p.4 line 6 of 2nd_recised_manuscript.pdf)

However, the fundamental interaction of the assembly formation; protein-protein interactions are difficult to regulate with small molecules for several reasons. **First**, large surface areas are usually involved in interactions between proteins, whereas the accessible surface areas of chemical ligands are small. **In addition**, there are generally no substantial grooves at the protein-protein interface, and there are few natural inhibitors of protein-protein interactions to guide ligand design³⁻⁵.

Cell adhesion by classical cadherin family proteins, calcium-dependent cell adhesive molecules, entails the protein assembly formation.

Comment from the reviewer

4. The font size used in Fig. 2c is still too small to read and will not reproduce well in production.

>Our response to the comment

We modified the figure accordingly. Please check the revised manuscript.

Other changes in the manuscript

(Original manuscript)

the driving forces that result in of Hit 1 binding appear to

(Revised p.10, line 11 of 2nd_recised_manuscript.pdf)

the driving forces that result in binding of Hit 1 appear to

(Original manuscript)

in the absence Hit 1

(Revised p.23, line 11 of 2nd_recised_manuscript.pdf)

In the absence of Hit 1

(Original manuscript)

cell aggregates are be too stable

(Revised p.31, line 9 of 2nd_recised_manuscript.pdf)

cell aggregates are too stable

(Original manuscript)

One tryptamine derivatives, serotonin, is

(Revised p.32, line 6 of 2nd_recised_manuscript.pdf)

One tryptamine derivative, serotonin, is

(Label of vertical axis in Fig. 3a)

ΔD (%)

(Revised)

D (%)

Reviewers' comments:

See attached PDF

1.
 - In my first round of comments, I mentioned that SEC-MALS is missing
 - The authors responded with a SEC instead that showed no difference between the monomer and dimer
 - Now the authors argue that their dilution was incorrect
 - I do not understand how the dilution cannot be corrected or how the dilution would affect the results (other than extremely high concentrations might result in aggregation, but this is not the case here)
2.
 - I disagree that sample injection into SEC-MALS (but they did not do SEC-MALS, they ran the SEC) dilutes the protein
 - I assume they mean that the sample elutes diluted
 - So now they argue that upon injection the dimer dissociates into a monomer due to the dilution
 - This seems unlikely and if it was the case, then the authors could never isolate the dimer but would always obtain a mixture of dimers and monomers, even at high concentration
 - In any case, their Figure 3e shows good SEC-MALS data for wild type X dimer and K14A mutant protein so the requested SEC-MALS experiment seems doable
3.
 - The rewording of the electron density identification does not seem to address the question
4.
 - The new electron density map Figure that says it is now a 2mFo-DFc does not seem to have such Fourier coefficients
5.
 - Tyr-140 engages in crystal contacts whereby its CE1 is 4 Å from symmetry-related Ala-5 CB

- Furthermore, that region is occupied by unaccounted electron density in PDB entry 4zmz
6.
 - I am sorry but the addition of “*arginine is inserted at the N termini*” and “*inserted methionine at the N termini*” remains cryptic to me
 7.
 - Now we learn that EDTA prevents dimer to monomer transition
 - The Figure R1 would be more helpful with an SDS-PAGE of the fractions
 - All protein chemistry problems add to the notion that the results might be overinterpreted and more complicated than they seem
 8.
 - PDB entry 4zmz has 91 water molecules and a resolution of 2.05 Å (space group C 1 2 1 as well)
 - Their structure is of a little lower resolution (2.3 Å) but one would still expect a similar number of water molecules
 - Did the authors check the water molecules of 4zmz and if these are present in their structure?
 - The table should please show the temperature factors for the ligands separately from the temperature factors of the protein

Response to the referees' comments and revisions that have been made

We thank the editorial board members and reviewer 1 for reviewing our manuscript thoroughly. We revised the manuscript in light of the comments. Please check the following point-by-point response to the reviewers' comments as well as the attached revised files ("3rd_Revised_manuscript.pdf", and "3rd_Revised_SI.pdf"). We also attached copies of the revised files showing track changes for the benefit of the reviewers. We hope these answers would resolve all the reviewers' concerns.

(black: comments from the reviewers, blue: our comments, red: inserted/added/modified sentences/figures/tables)

Reviewer 1

Original comments from Reviewer 1

1.

- In my first round of comments, I mentioned that SEC-MALS is missing
- The authors responded with a SEC instead that showed no difference between the monomer and dimer
- Now the authors argue that their dilution was incorrect
- I do not understand how the dilution cannot be corrected or how the dilution would affect the results (other than extremely high concentrations might result in aggregation, but this is not the case here)

2.

- I disagree that sample injection into SEC-MALS (but they did not do SEC-MALS, they ran the SEC) dilutes the protein
- I assume they mean that the sample elutes diluted
- So now they argue that upon injection the dimer dissociates into a monomer due to the dilution
- This seems unlikely and if it was the case, then the authors could never isolate the dimer but would always obtain a mixture of dimers and monomers, even at high concentration
- In any case, their Figure 3e shows good SEC-MALS data for wild type X dimer and K14A mutant protein so the requested SEC-MALS experiment seems doable

3.

- The rewording of the electron density identification does not seem to address the question

4.

- The new electron density map Figure that says it is now a 2mFo-DFc does not seem to have such Fourier coefficients

5.

- Tyr-140 engages in crystal contacts whereby its CE1 is 4 Å from symmetry-related Ala-5
- Furthermore, that region is occupied by unaccounted electron density in PDB entry 4zm

6.

- I am sorry but the addition of “arginine is inserted at the N termini” and “inserted methionine at the N termini” remains cryptic to me

7.

- Now we learn that EDTA prevents dimer to monomer transition
- The Figure R1 would be more helpful with an SDS-PAGE of the fractions

- All protein chemistry problems add to the notion that the results might be overinterpreted and more complicated than they seem

8.

- PDB entry 4zmz has 91 water molecules and a resolution of 2.05 Å (space group C 1 2 1 as well)

- Their structure is of a little lower resolution (2.3 Å) but one would still expect a similar number of water molecules

- Did the authors check the water molecules of 4zmz and if these are present in their structure?

- The table should please show the temperature factors for the ligands separately from the temperature factors of the protein

Point-by-point response to the comments from Reviewer 1

We thank the reviewer for the criticisms. Following the instructions from the editor, we provide answers to the comments No.3 to No. 8 as follows.

Comment from the reviewer

3.

- The rewording of the electron density identification does not seem to address the question

>Our response to the comment

We have carefully read and considered the previous (second review) comment again. We seriously answer your question “how do we know this density is not from the crystallization solution”. In our refinement process regarding this crystal structure, we wondered if the density was derived from some of the crystallization reagents such as PEG400 or DMSO, but the density was so large and clear that none of those reagents seemed fitting the density. Above all, the shape of the relevant electron density was most reasonably explained by Hit 1. After refinement, no difference Fourier was observed. Therefore, it would be most reasonable to think that the relevant electron density is derived from Hit 1. Also, as mentioned in the manuscript, the binding of Hit 1 at the position was validated together with the mutation work in SPR.

Comment from the reviewer

4.

- The new electron density map Figure that says it is now a 2mFo-DFc does not seem to have such Fourier coefficients

>Our response to the comment

This 2mFo-DFc is shaved for the region of Hit 1; this way of drawing a map is routinely adopted in figures from other crystallographers too. We have checked Figure 2 and Supplementary Figure S2 again, to find the electron density of Figure 2 was still at 1.0 σ , although the caption said it was 0.9 σ . We have corrected it and now we confirmed that Figure 2(b) and Supplementary Figure S2 have correct Fourier coefficients with correct σ value.

(Original Figure 2)

(New Figure 2)

Comment from the reviewer

5.

- Tyr-140 engages in crystal contacts whereby its CE1 is 4 Å from symmetry-related Ala-5
- Furthermore, that region is occupied by unaccounted electron density in PDB entry 4zmz

>Our response to the comment

We had noticed the interaction between Y140 and symmetry-related A5 from the beginning. Including this interaction, the only difference in the conditions between 4zmz and this structure was whether or not Hit 1 (and DMSO) was added or not. The unaccounted electron density around the region was too weak to model as a water molecule, leaving little possibility that the electron density rather than Hit 1 caused the difference in the sidechain of Y140. Therefore, we decided to leave the relevant sentences as they are.

Comment from the reviewer

6.

- I am sorry but the addition of “arginine is inserted at the N termini” and “inserted methionine at the N termini” remains cryptic to me

>Our response to the comment

We have modified grammatical mistakes. Also, for the benefit of readers, we added Supplementary Table S1 as a list of construct and sequence.

(Supplementary Table S1)

Supplementary Table S1 | Cadherin constructs used in this study. ^a

Construct name ^b	Oligomeric state	Sequence	Mutation
EC12	Strand-swap dimer	1-241	No mutation
MEC12	X dimer	Met + 1-241	Met 0
REC12	monomer	Arg + 1-241	Arg 0
C-terminal-deleted MEC12	X dimer	Met + 1-213	Met 0
C-terminal-deleted REC12	monomer	Arg + 1-213	Arg 0

^aThis list does not include a full-length construct, or tag information used for SPR, heliX, and liposome aggregation assay.

^bM and R means the one-letter code of the additional amino acid (methionine and arginine) inserted at the N-terminus of EC12.

Comment from the reviewer

7.

- Now we learn that EDTA prevents dimer to monomer transition
- The Figure R1 would be more helpful with an SDS-PAGE of the fractions
- All protein chemistry problems add to the notion that the results might be overinterpreted and more complicated than they seem

>Our response to the comment

We are now showing you the SDS-PAGE for **Fig. R1 (b)**, which was performed on the same day of SEC of **Fig. R1 (b)**. We did not perform SDS-PAGE analysis for the experiment described by **Fig. R1 (a)**, since the experiment was conducted using the already purified EC12 construct.

Figure | SDS-PAGE analysis corresponding to Fig.R1(b) in 2nd revision. Since the peak that eluted around 100-140 mL was obviously void peak, the small peak shown by a red square was analyzed with SDS-PAGE (CBB stained). The molecular weight of this peak was approximately 17 kDa, which is the value for SUMO-tag, not the P-cadherin construct. In this SDS-PAGE, protein solution of Ulp1-treated His-SUMO-EC12, which yielded EC12 and His-SUMO-tag and Ulp1 (and a small amount of uncut His-SUMO-EC12), was used as a control sample.

Anyway, not to overinterpret the results from the liposome aggregation assay described in the manuscript, we modified the relevant sentences as follows. Especially, we decided not to use the word “kinetics,” since the direct result to show the modulated kinetics of X dimerization has not been provided.

(Original manuscript)

The fragments identified herein had an impact on kinetics; probably by trapping X dimer to an incomplete, metastable state, although whether Hit 1 modulates on-rate or off-rate of X dimerization remains unclear.

(Revised p. 30 line 13 of 3rd_revised_manuscript.pdf)

The fragments identified herein had an impact on **the dimerization process**; probably by trapping X dimer to **another state that lacks a key hydrogen bond**.

(Original manuscript)

SPR-based compound screen can provide an ideal platform to select such a kinetic modulator

(Revised p. 30 line 14 of 3rd_revised_manuscript.pdf)

SPR-based compound screen can provide an ideal platform to select such a **dimerization** modulator

(Original manuscript)

a slight, kinetic level inhibition of the key step can lead to significant inhibition of formation of the final state.

(Revised p.31 line 4 of 3rd_revised_manuscript.pdf)

a slight inhibition of the key step can lead to significant inhibition of formation of the final state.

(Original manuscript)

our strategy for kinetic regulation of protein-protein interactions

(Revised p.31 line 17 of 3rd_revised_manuscript.pdf)

our regulation strategy of protein-protein interactions

Comment from the reviewer

8.

- PDB entry 4zmz has 91 water molecules and a resolution of 2.05 Å (space group C 1 2 1 as well)
- Their structure is of a little lower resolution (2.3 Å) but one would still expect a similar number of water molecules
- Did the authors check the water molecules of 4zmz and if these are present in their structure?
- The table should please show the temperature factors for the ligands separately from the

temperature factors of the protein

>Our response to the comment

We checked whether there was any electron density that can be modeled as water molecules manually and thoroughly.

We show the temperature factors for the ligands separately from that of the proteins. Please check the revised manuscript.